

**In situ measurements of cloud microphysics and aerosol over**
**coastal Antarctica during the MAC campaign**
**Sebastian J. O'Shea[1], Thomas W. Choularton[1], Michael Flynn[1], Keith N. Bower[1],**
**Martin Gallagher[1], Jonathan Crosier[1,2], Ian Crawford[3], Zoë Fleming[4], Constantino**
**Listowski[4]\*, Amélie Kirchgaessner[4], Russell S. Ladkin[4], and Thomas Lachlan-Cope[4]**
[1]{School of Earth and Environmental Sciences, University of Manchester, Oxford Road,
Manchester, M13 9PL, UK}
[2]{National Centre for Atmospheric Science, University of Manchester, Oxford Road,
Manchester, M13 9PL, UK}
[3]{National Centre for Atmospheric Science, Department of Chemistry, University of
Leicester, Leicester, LE1 7RH, UK}
[4]{British Antarctic Survey, NERC, High Cross, Madingley Rd, Cambridge CB3 0ET, UK}
\*now at: LATMOS/IPSL, UVSQ Université Paris-Saclay, UPMC Univ. Paris 06, CNRS,
Guyancourt, France
Correspondence to: S. J. O'Shea (sebastian.oshea@manchester.ac.uk)
**Abstract**
During austral summer 2015 the Microphysics of Antarctic Clouds (MAC) field campaign
collected detailed airborne and ground based in situ measurements of cloud and aerosol
properties over coastal Antarctica and the Weddell Sea. This paper presents the first results
from the experiment and discusses the key processes important in this region.
The sampling was predominantly of stratus cloud, at temperatures between -20 and 0 °C.
These clouds were dominated by supercooled liquid water droplets, which had a median
concentration of 113 cm$^{-3}$ and an inter-quartile range of 86 cm$^{-3}$. The concentration of large
aerosols (0.5 to 1.6 μm) decreased with altitude and were depleted in airmasses that originated





over the Antarctic Continent compared to those more heavily influenced by the Southern
Ocean and sea ice regions. The dominant aerosol in the region was hygroscopic in nature,
with the hygroscopicity parameter, κ having a median value for the campaign of 0.64
(interquartile range = 0.34). This is consistent with other remote marine locations that are
dominated by sea salt/sulphate.
Cloud ice particle concentrations were highly variable with the ice tending to occur in small
isolated patches. Below ca 2000 m glaciated cloud regions were more common at higher
temperatures; however the clouds were still predominantly liquid throughout. When ice was
present at temperatures higher than -10 °C, secondary ice production most likely through the
Hallet-Mossop mechanism lead to ice concentrations 1 to 3 orders of magnitude higher than
the number predicted by commonly used primary ice nucleation parameterisations. The
drivers of the ice crystal variability are investigated. No clear dependence on the droplet size
distribution was found. However, higher ice concentrations were found in updrafts and
downdrafts compared to quiescent regions. The source of first ice in the clouds remains
uncertain, but may include contributions from biogenic particles, blowing snow or other
surface ice production mechanisms.

## 1 Introduction

Antarctic clouds have a central role in the weather and climate at high southern latitudes
(Lubin et al., 1998; Lawson and Gettelman, 2014). Through snow precipitation and their
radiative effects they are key to the mass balance of the Antarctic ice sheet, which impacts on
global sea levels (van den Broeke et al., 2011) and Southern Ocean circulation (Bromwich et
al., 2012). In addition it has been suggested that changes in Antarctic clouds can influence
weather patterns as far away as the tropics and even the extratropics of the Northern
Hemisphere (Lubin et al., 1998).
Despite their importance Antarctic clouds are some of the least studied of any region around
the globe (Bromwich et al., 2012). The remote location and harsh conditions cause significant
logistical challenges for field projects in this region. As a consequence there is evidence that
clouds and their radiative properties are poorly represented in weather and climate models
over Antarctica (Bromwich et al., 2013; King et al., 2015) and the Southern Ocean (Bodas-
Salcedo et al., 2012; 2016).



Key uncertainties concern the aerosol in the region, in particular the number and sources of
cloud condensation nuclei (CCN) and ice nucleating particles (INPs). Conventional
parameterisations predicting INP concentrations have primarily been developed using
measurements at mid-latitudes (e.g. Cooper, 1986; DeMott et al., 2010) and may not be
appropriate for Antarctica.  A number of intensive field campaigns have been conducted
studying Arctic clouds (McFarquhar and Cober, 2004; McFarquhar et al., 2007; Verlinde et
al., 2007; Lloyd et al., 2015a), however analogies between the polar regions may also not be
appropriate. The Arctic receives significant anthropogenic aerosol input due to its proximity
to industrial nations, and is therefore likely to have significantly different type and number of
CCN/INP (Mauritsen et al., 2011; Lathem et al., 2013; Liu et al., 2015).
To date, Antarctic INP measurements have mostly been made at surface sites. Measurements
of snowflake residuals at the South Pole identified the long range transport of clays as the
likely dominant source (Kumai, 1976). However, interpretation of these measurements is
complicated due to secondary aerosol scavenging by the snowflakes and precipitation,
meaning they contain particles in addition to the original nuclei. More recently, filter samples
at the South Pole detected INPs that were active between -18 and -27°C, with concentrations
of 1 $L^{-1}$ at -23 °C. Mineral dusts transported from the Patagonian deserts were identified as
the likely source (Ardon-Dryer et al., 2011).  A synthesis of INP measurements prior to 1988
from the high southern latitudes (> 60°S), found mean concentrations between $2x10^{-4}$ and 0.2
$L^{-1}$ at -15°C (Bigg, 1990). Given the general absence of other local INP sources, biogenic
INPs may have a more important role in the Antarctic than in other regions. Biological
species (pollen, bacteria, fungal spores and plankton) have been shown to act as INP at
significantly higher temperatures than mineral dusts (> -15°C) (Möhler et al., 2007; Alpert et
al., 2011; Murray et al., 2012; Amato et al., 2015; Wilson et al., 2015). However, Antarctic
snowfall has been shown to be relatively depleted of biological INP (Christner et al., 2008)
and bacteria commonly found in sea ice may not be effective INP (Junge and Swanson,
2007). The few in situ measurements of Antarctic clouds to date have suggested the
importance of secondary ice processes (Grosvenor et al., 2012; Lachlan-Cope et al., 2016).
There is a clear need for more direct measurements to test and improve the representation of
Antarctic clouds in climate/weather models. This paper presents both ground based and
airborne measurements of cloud and aerosol properties during the 2015 Microphysics of
Antarctic Clouds (MAC) field campaign aimed at addressing this. Section 2 provides an





overview of the campaign and the measurement techniques used. Section 3 presents a
statistical overview of the aerosol and cloud observations using all available measurements.
Section 4 discusses the key microphysical processes. Conclusions are presented in Sect. 5.
**2   Methods**
**2.1   Campaign and meteorological overview**
The MAC experiment comprised both airborne and ground based measurements of cloud and
aerosol properties. Ground based measurements were performed at the Clean Air Sector
Laboratory (CASLab), which is located at the Halley research station. Halley is a coastal
Antarctic base on the Brunt Ice shelf, approximately 30 km from the Weddell Sea (75.6° S,
26.7° W). The CASLab is located 1 km south of the main Halley buildings and receives
minimal pollution from the base and vehicle traffic due to the prevailing easterly wind (Jones
et al., 2008). All CASLab measurements were filtered using the wind direction to help
remove any remaining influence from the base.
The airborne measurements were collected using the British Antarctic Survey's Twin Otter
MASIN research aircraft (King et al., 2008). Twenty-four flights (a total of 80 hours) were
performed during November and December 2015 from Halley. These flights have the nominal
flight numbers 212 to 235. The flights were predominantly performed over the Weddell Sea
(see Fig. 1), which at this time and location was covered by a mixture of broken sea ice and
polynyas. This is shown in Fig. 1 together with the sea ice fraction from the Nimbus-7
Multichannel Microwave Radiometer (SMMR) and Defense Meteorological Satellite Program
(DMSP) SSM/I-SSMIS passive microwave data (Cavalieri et al., 1996.) One flight sampled
clouds in-land over the Antarctic continent (Flight 233). In addition a transit took place from
Rothera research station on the Antarctic Peninsula (Flights 212 to 215); however not all
instruments were available during these transit flights. Since the aircraft was not pressurised,
the measurements were restricted to altitudes below approximately 4000 m. As a
consequence, the majority of clouds were sampled over the temperature range -11 and -3 °C
(79%). Seventeen percent of in-cloud measurements were collected at temperatures below -11
°C and 4% at temperatures higher than -3 °C. In total 17 hours of sampling during the
campaign was performed in-cloud.



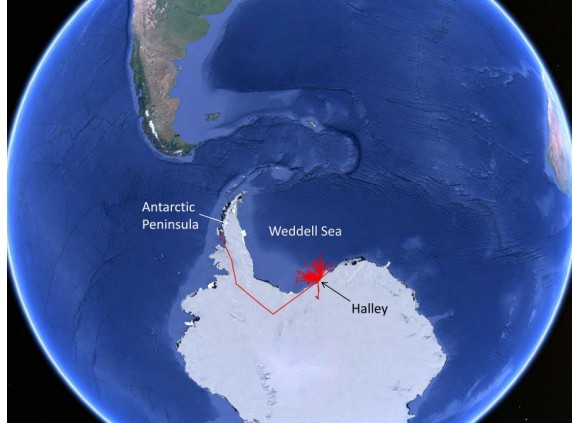

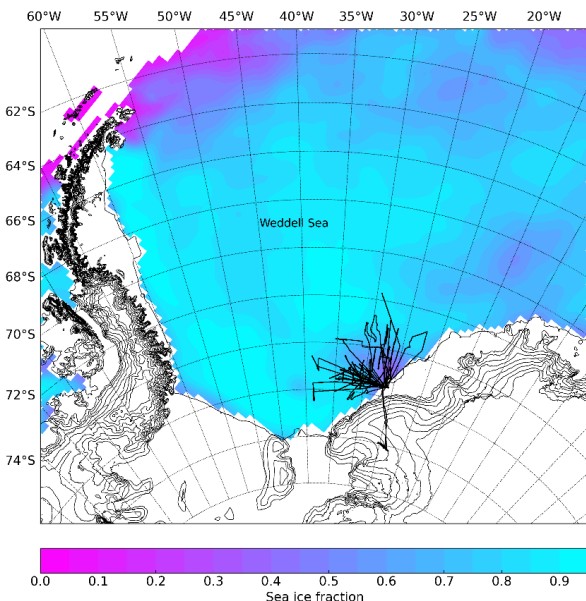

4    *Figure 1. Top panel: Flight tracks during the MAC field project (source Google Earth).*

5    *Lower panel: shows the sea ice fraction on the Weddell Sea (Cavalieri et al., 1996) during the*

6    *experimental period.*



The clouds sampled were generally stratiform. The exception to this was Flight 224, which
sampled frontal clouds. Back trajectory analysis showed that two broad regimes were present
during the project. The earlier flights (up to Flight 223) generally sampled airmasses that had
travelled south over the Southern Ocean and Weddell Sea. Later in the campaign there was a
transition to airmasses with greater influence from the Antarctic continent.
**2.2    Aircraft**
During MAC the Twin Otter MASIN research aircraft was fitted with a range of in situ
aerosol and cloud microphysical instrumentation. Cloud particle size distributions were
derived using the images from two optical array probes (OAP): a 2DS (2D-stereo, SPEC Inc.,
USA, see Lawson et al., 2006) with a nominal size range of 10 to 1280 μm (10 μm pixel
resolution) and a CIP-25 (Cloud Imaging Probe, DMT Inc., USA, Baumgardner et al., 2001)
with a size range of 25 to 1600 μm (25 μm pixel resolution).
Particle size distributions over the size range from 0.5 to 50 μm were recorded using a Cloud
Aerosol Spectrometer (CAS, DMT Inc., USA, Baumgardner et al., 2001).  The CAS sizing
was calibrated by the manufacturer using polystyrene latex (PSL) spheres ($< 2$ μm) and glass
beads ($> 2$ μm) (Baumgardner et al., 2014). During MAC the sizing of the CAS's larger bins
($>10$ μm) was also validated using reference glass calibration beads and show little instrument
drift (see Fig 2.).
The aircraft was also fitted with a Cloud Droplet Probe (CDP-100, DMT Inc.) for observing
cloud droplets between 3 and 50 μm (Lance et al., 2010). Following the method detailed by
Rosenberg et al. (2012), glass beads were used to determine the CDP's size bin centres and
widths. The 2DS, CIP-25 and CAS were fitted with anti-shatter tips to minimise ice break-up
on their leading edges (Korolev et al., 2011). For full details of the data processing and
quality control of the 2DS and CIP-25 measurements see Crosier et al. (2011) and Taylor et
al. (2016). It should be noted that in addition to the use of anti-shatter tips, an inter-arrival
time algorithm was used to further reduce shattering artefacts on the 2DS and CIP-25
datasets. Ice mass content was determined from the 2DS and CIP-25 images using the Brown
and Francis (1995) mass-diameter relationship. As an example Fig. 2 shows a comparison
between the CDP, CAS, 2DS, and CIP-25 size distributions for Flight 227. Unless stated
otherwise all flight data presented has been averaged to 10 second intervals.





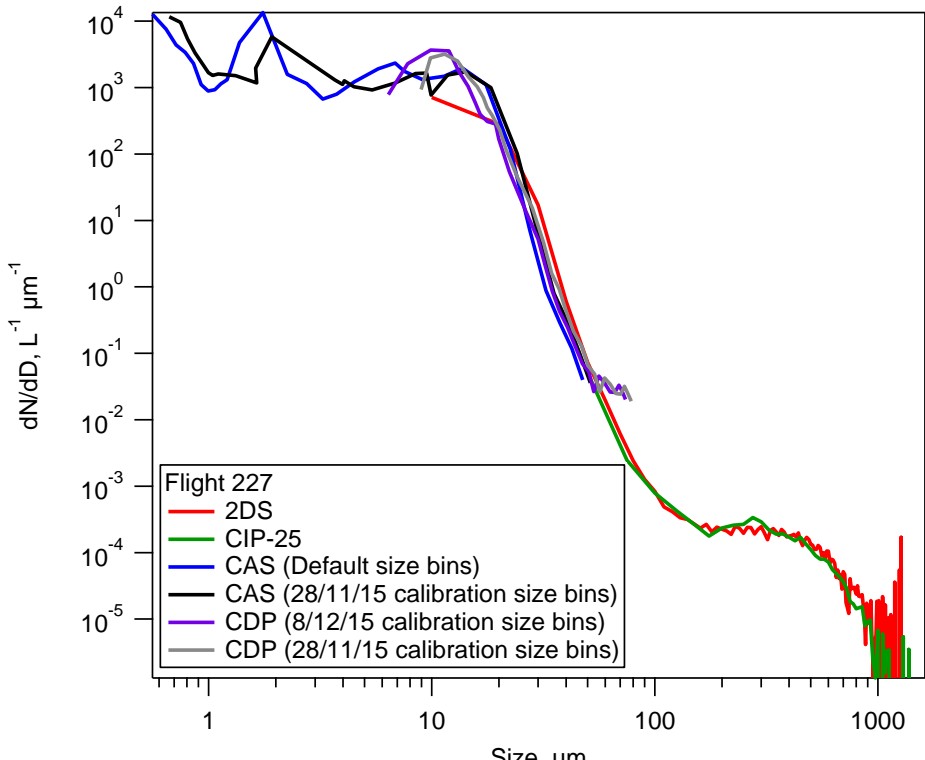

*Figure 2. Average size distribution for Flight 227 comparing the 2DS, CIP-25 CDP and CAS*

*probes. The CAS and CDP shows the Flight 227 size distributions using results from the bead*

*calibrations performed during the campaign in order to monitor instrument performance.*

Following Crosier et al. (2011), 2DS images were classified based on a geometric analysis of

their circularity. Particles containing less than 50 pixels (equivalent to a diameter of

approximately 80 μm) were not classified since they contain insufficient pixels to accurately

determine their shape. Particles with circularity values less than 1.2 were classified as low

irregular (LI) and are indicative of liquid drops. Circularity values greater than 1.4 are

associated with ice crystals and are classified as high irregular (HI). Visual inspection of the

LI and HI images confirmed that they were almost all liquid droplets and ice crystals,

respectively. Circularities between 1.2 and 1.4 are classified as medium irregular (MI).

Interpretation of the MI category with respect to the particle phase is more ambiguous than

the other categories. In general, the MI images were of quasi-spherical ice crystals, such as





recently frozen drops, however they may also include some poorly imaged liquid drops that
should be classified as LI. During MAC the concentration of MI particles was generally
significantly less than HI particles. The mean ratio HI:MI for the campaign was 7 (see also
Sect. 3.1). This suggests that the HI concentration is likely a good proxy for the ice crystal
concentration. However to highlight the uncertainty in the phase separation, in Sect. 3 the MI
concentration is also shown along with the HI concentration.
Aerosol instrumentation on the aircraft included a GRIMM optical particle counter (GRIMM
Model 1.109) capable of detecting aerosol particles over the size range from 0.25 to 32 μm.
The GRIMM sampled through a Brechtel Model 1200 isokinetic aerosol inlet with a >95%
sampling efficiency for particles in the size range 0.01μm to 6 μm. Inlet losses only become
significant for particles >6 μm and here we only consider the concentration of particles below
2μm. Total aerosol concentrations of particles >10 nm in size were determined using a
Condensation Particle Counter (CPC, TSI Inc. Model 3772).
The aircraft was also fitted with instrumentation to measure temperature, turbulence,
humidity, radiation and surface temperature. See King et al. (2008) for full details.
**2.3   Ground site measurements**
Aerosol instrumentation was installed at the CASLab sampling from its central aerosol stack
(Jones et al., 2008) for the measurement period from 27 November 2015 to 15 December
2015. A Scanning Mobility Particle Sizer (SMPS, TSI) was used to generate a quasi-
monodisperse aerosol flow. The SMPS performed 27 discrete steps over the aerosol size
range from 30 to 500 nm. Downstream of the SMPS the flow ($1 \, L^{-1}$) was split isokinetically
between a cloud condensation nuclei counter (CCNc, Droplet Measurement Technology
Model CCN-100) and a condensation particle counter (CPC, TSI). The CCN concentration
was measured at super saturations of 0.05%, 0.13%, 0.20%, 0.26% and 0.34%. The activated
cloud droplet fraction was determined by the ratio of activated particles from the CCN to the
total number of particles measured by the CPC. The dry diameter at which 50% of particles
were activated ($D_{50}$) was determined by fitting a sigmoid curve to the activated fraction size
spectrum (Whitehead et al., 2016). The total CCN concentration was determined by
integrating the concentration of particles larger than $D_{50}$. The hygroscopicity parameter κ was
derived from κ-Köhler theory using the $D_{50}$ and supersaturation values (Petters and
Kreidenweis, 2007).





The SMPS and CCNc were calibrated at the beginning and end of the campaign (Good et al.,
2010). The SMPS was size calibrated using NIST traceable polystyrene latex spheres (PSLs).
Ammonium sulphate and sodium chloride were used to calibrate the CCNc supersaturations,
by comparing measured values to theoretical ones from the Aerosol Diameter Dependent
Equilibrium Model (ADDEM) (Topping et al., 2005).
Additional measurements were provided by an Aerodynamic Particle Sizer (TSI Model 3321)
which provided aerodynamic particle size concentration measurements over the size range
$0.5<D<20$ µm and in the size range $0.3<D<20$ µm from simultaneous aerosol scattering cross
section measurements.
Continuous measurements of airborne bio-fluorescent particle concentrations (primary
biological and mixed biological and non-biological) were also made at CASLab using a
Wideband Integrated Bioaerosol Spectrometer (WIBS Model Dstl-3, Gabey et al. 2010,
Crawford et al. 2014, 2015). Measurements from this instrument will be described in detail in
a separate paper.
**3    Results**
**3.1    Cloud microphysics**
The following section presents a broad overview of the microphysical measurements during
the MAC field campaign. For this analysis "in-cloud" measurements were determined as
periods when the liquid water content (LWC) was greater than 0.01 g m$^{-3}$ or when particles
were detected by the 2DS. Flight 224 is excluded from this bulk analysis since this flight
sampled frontal cloud, while the other flights sampled shallow layer cloud. The ice mass
fraction (IMF) is calculated as the ratio of the ice mass to the total condensed water. Here the
ice mass is taken as the sum of the HI and MI 2DS categories, while the liquid mass is taken
as the sum of the CAS droplets (>3 µm) and the 2DS LI category. Ice mass fractions of 0 and
1 represent fully liquid and glaciated conditions, respectively.  Figure 3 (black line) shows the
frequency distribution of ice mass fraction based on all 1 Hz measurements in layer clouds
sampled during MAC. As can be seen in Fig. 3 the clouds were dominated by liquid water.
Ice mass fractions between 0 and 0.1 were observed 90% of the time, while only 6% of cases
had values between 0.9 and 1. Figure 4 shows the ice mass fraction as a function of height,
the black line shows the mean for each altitude bin. For altitudes below ca. 2000 m there is a



general trend of glaciated conditions becoming more prevalent with decreasing altitude (and
increasing temperature). At temperatures higher than -3 °C glaciated conditions (IMF greater
than 0.9) were responsible for 15% of observations, compared to 7% at temperatures between
-8 and -3 °C. Above 2000m glaciated regions become more frequent with increasing altitude,
however this is based on comparatively few observations.
Measurements in Arctic stratus/stratocumulus generally find these clouds to be similarly
dominated by liquid drops (McFarquhar and Cober, 2004; McFarquhar et al., 2007; Lloyd et
al., 2015a). McFarquhar et al. (2007) also show a trend of increasing IMF with increasing
distance from cloud top (and increasing temperature) during the Mixed-Phase Arctic Cloud
Experiment (M-PACE). Glaciated conditions were observed during 23% of their
measurements. This is significantly more than during MAC, possibly due to lower INP
concentrations available for primary ice development in the Antarctic compared to the Arctic.
Flight 224 sampled cloud layers at the rear of an occluded front that was associated with a
low pressure system north of Halley. Several layers were observed between -19 °C and -1 °C
with ice crystals precipitating between the layers. As shown in Fig. 3 (red line) ice was more
frequently observed in these clouds than during the flights where stratocumulus/stratus clouds
were sampled. Twenty-four percent of measurements had ice mass fractions between 0.9 and
1, while 32% of observed ice mass fraction values were between 0.1 and 0.9. Droplet number
concentrations were comparatively low with a mean of 40 (29 at 1$\sigma$) cm$^{-3}$.





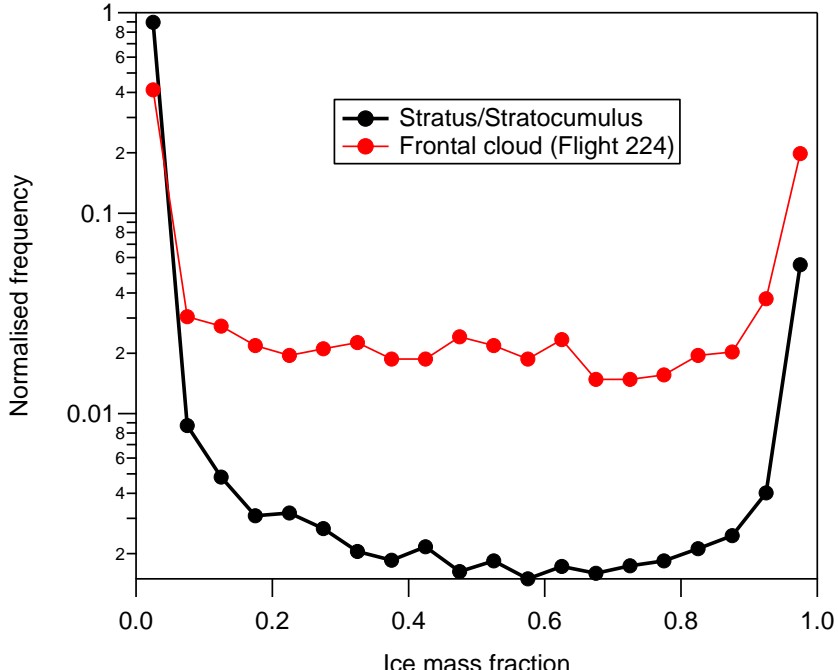

2    *Figure 3. Frequency distribution of the 1 Hz cloud ice mass fraction measurements.*





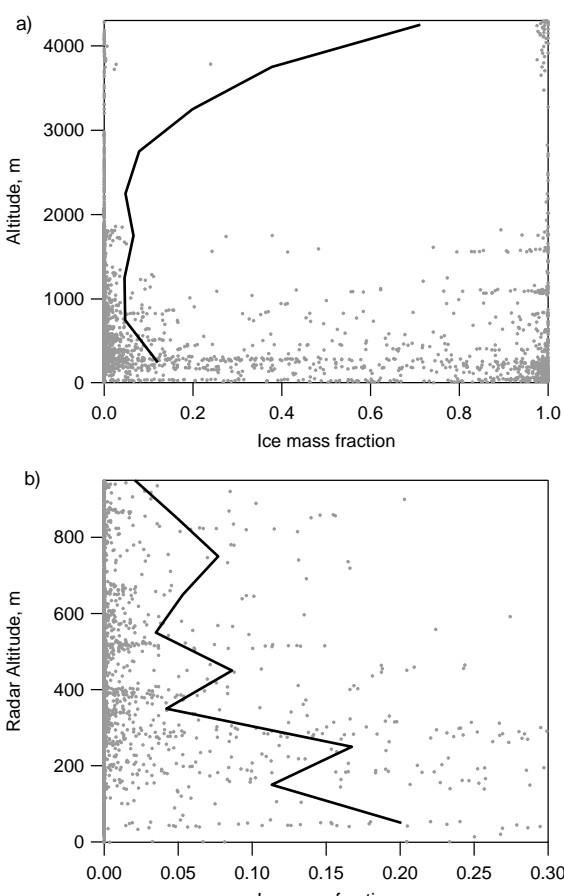

Figure 4. Ice mass fraction as a function of altitude, black lines show the average ice mass fraction for each altitude bin.

The droplet number concentration as a function of temperature is shown in Fig. 5a. This was found to be relatively consistent and temperature independent during the campaign with a median of 113 cm$^{-3}$ and an inter-quartile range of 86 cm$^{-3}$. An exception to this is Flight 217, when anomalously high droplet concentrations were observed at -23 °C (mean 310 cm$^{-3}$). The 2DS was not available during this flight but the CIP observations suggest that ice was not present in this cloud. The reason for the enhanced droplet concentrations is not clear, however the aerosol concentrations below the cloud layer was similarly elevated with the CPC recording concentrations of over 1200 scm$^{-3}$, compared to the median for the campaign of 408





scm$^{-3}$. Back trajectory analysis showed that in the previous days this airmass travelled over
the Southern Ocean from South America.
The cloud droplet concentrations during MAC are found to be comparable with previous
observations from the Antarctic Peninsula (Lachlan-Cope et al., 2016) and also Arctic
summer stratocumulus (Lloyd et al., 2015a). Droplet concentrations over the Antarctic
Peninsula varied between 60 and 200 cm$^{-3}$ (Lachlan-Cope et al., 2016). Concentrations on the
eastern side of the Peninsula were moderately higher than on the west, which may be due to
the greater sea ice coverage on the eastern side. It has been suggested that sea ice may provide
a more efficient source of sea-salt aerosol, and therefore CCN, than open waters (Yang et al.,
2008). Recent measurements and modelling found that sea ice made a significant contribution
to the winter sea-salt aerosol loading at coastal (Dumont d'Urville) and central (Concordia)
East Antarctic sites (Legrand et al., 2016).
The number of highly irregular particles observed by the 2DS can be used as a proxy for the
number of ice crystals; it is shown as a function of temperature in Fig. 5b. Box and whisker
plots show statistics for those regions of the cloud where ice is present (i.e. excluding regions
with only liquid cloud water). The 2DS was not operated during the flights previous to flight
218 so measurements are only available at temperatures higher than -20 °C. The two lowest
temperature bins in Fig. 5b show the highest concentration of ice crystals. However these
measurements come from only one flight (Flight 226) where the base (4000 m) of high cloud
was sampled. These crystals (predominantly rosettes and aggregates) are highly likely to have
been nucleated at lower temperatures higher up in the cloud which then sedimented down to
be sampled by the aircraft. Above -15 °C there is a trend of the ice crystal concentrations
showing greater variability and higher median concentrations with increasing temperature. Ice
in the clouds tended to occur in small patches. A histogram of the spatial extent of ice patches
shows that they increase in frequency with decreasing length up to the maximum resolvable
by the 2DS measurements (a sampling frequency of 10s corresponds to a spatial scale of ca
600m).





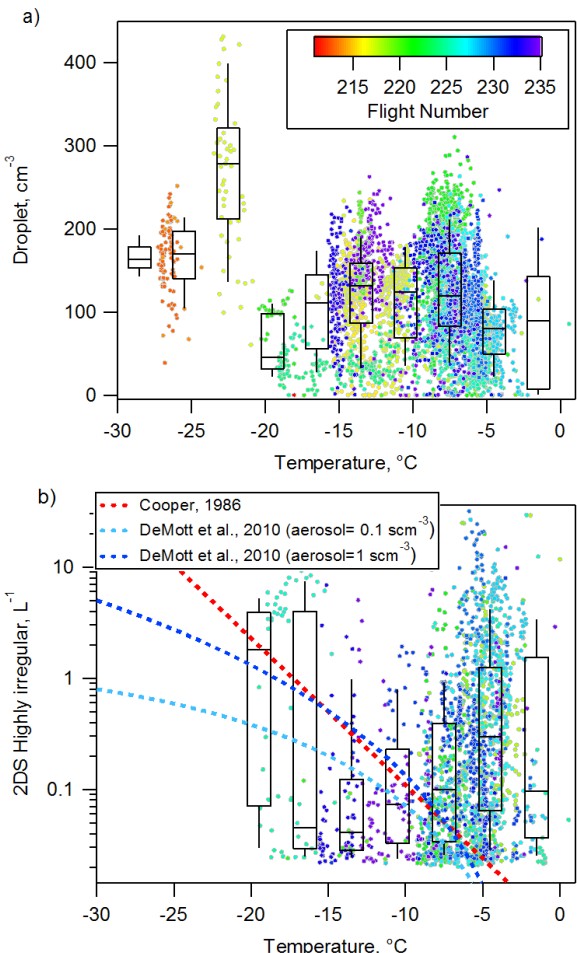

*Figure 5. Box and whisker plots summarising in cloud measurements (averaged over 10 s) as a function of temperature. Plate a) shows the concentration of cloud droplets ($cm^{-3}$), measured by CAS, while b) shows the concentration of ice particles measured by 2DS, based on those classified as highly irregular (see text for details). The concentration of ice nucleating particles predicted by the DeMott et al. (2010) parameterisation with a high (1 $scm^3$) and low (0.1 $scm^3$) aerosol input are shown as dark and light blue lines, respectively in b). The red line is the predicted ice particle concentration according to the Cooper (1986) parameterisation.*



Previous observations of Arctic mixed phase clouds found that the presence of precipitating
ice particles (> 400 μm) was associated with the number of large drops (>30 μm), however
the precise nucleation mechanism through which this occurs is uncertain (Lance et al., 2011).
To identify if a similar relationship was present during MAC Fig. 6a shows the relationship
between the 2DS HI and the 2DS LI particles (droplets larger than approximately 80 μm) over
the temperature range -8 to -3 °C. Figures 6b and 6c show similar plots for the CAS
measurements of droplets larger than 30 and 20 μm, respectively. The HI concentrations are
binned based on the droplet concentration and the 25, 50 and 75 percentiles are shown as
black lines. When examining statistics for all stratus flights we find no evidence that the ice
concentrations increase due to the presence of large drops. However, any relationship may be
obscured as drops are depleted by ice crystal growth through riming and the Wegener-
Bergeron-Findeisen process.

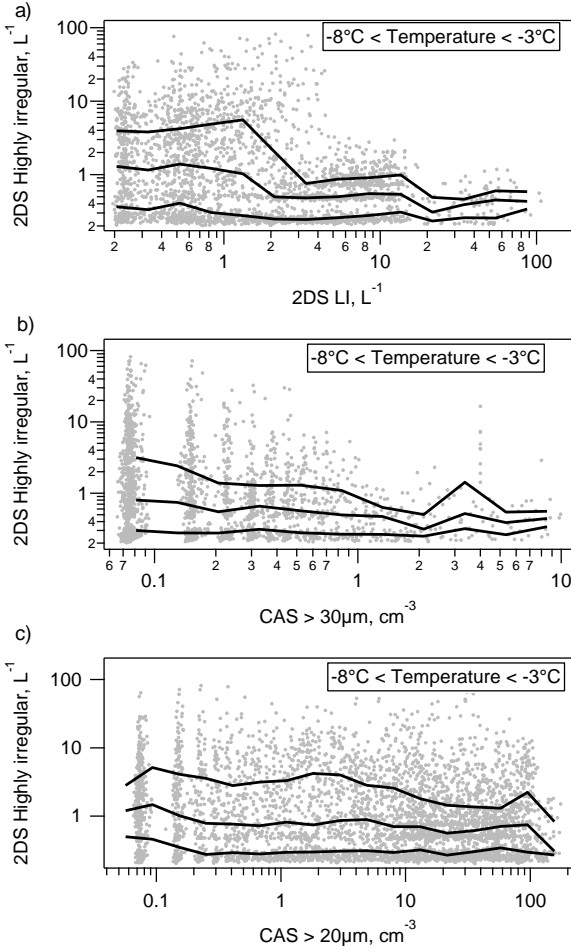

*Figure 6a. The relationship between the concentration of highly irregular (2DS HI) particles*

*and low irregular particles (2DS LI) (low irregular particles greater than approximately 80*

*μm) for the temperature range -8 to -3 °C. Figures 6b and 6c show the relationship with the*

*concentration of droplets larger than 30 and 20 μm, respectively. The black lines are the 25th,*

*50th and 75th percentile of the 2DS HI concentration for each droplet concentration bin.*

Similar results are found when case studies for individual flights are examined. Figure 7a
shows a comparison between the particle size distributions for three periods with quite
different degrees of glaciation during a constant altitude run at -5 °C during Flight 218. Time
series of the microphysical properties during this run are shown in Fig. 8. During this run



there were patches of ice with concentrations of several per litre and regions where no ice was
present. However, there are no distinct differences in the droplet spectrum for these three
cases. Figure 7b shows a similar plot for a constant altitude run at -6 °C during Flight 219.
During times with very high ice concentrations (2DS HI up to 50L$^{-1}$, blue line) the droplets
are depleted compared to the cases when the 2DS HI concentration was 1 L$^{-1}$ and 0 L$^{-1}$.

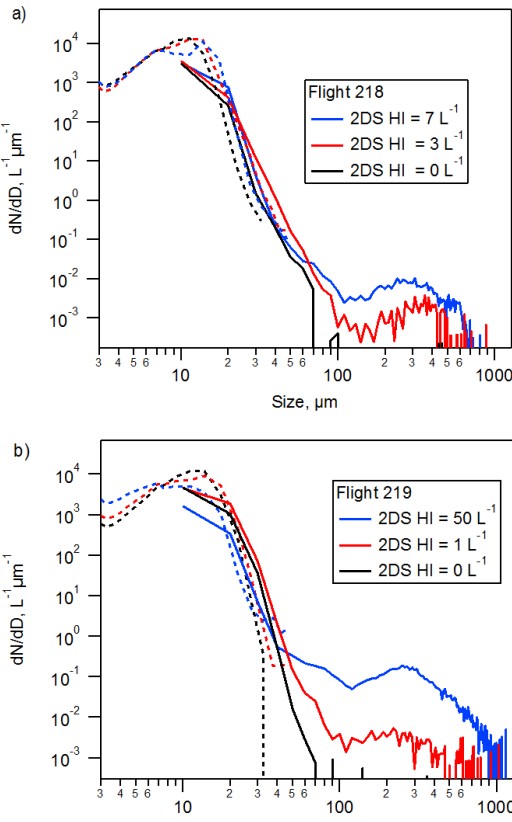

*Figure 7a. Comparison between the size distributions for 3 regions sampled in the constant*
*altitude run at -5 °C during Flight 218, these are where the concentration of highly irregular*
*particles (2DS HI) was 7 L$^{-1}$ (4:04 GMT), 3 L$^{-1}$ (3:58 GMT) and 0 L$^{-1}$ (3:52 GMT). Time*
*series of the microphysical measurements during this run are shown in Figure 8. Figure 7b*
*shows a similar plot for a run at -6 °C during Flight 219 when the 2DS highly irregular*
*concentration was 50 L$^{-1}$, 1 L$^{-1}$ and 0 L$^{-1}$.*





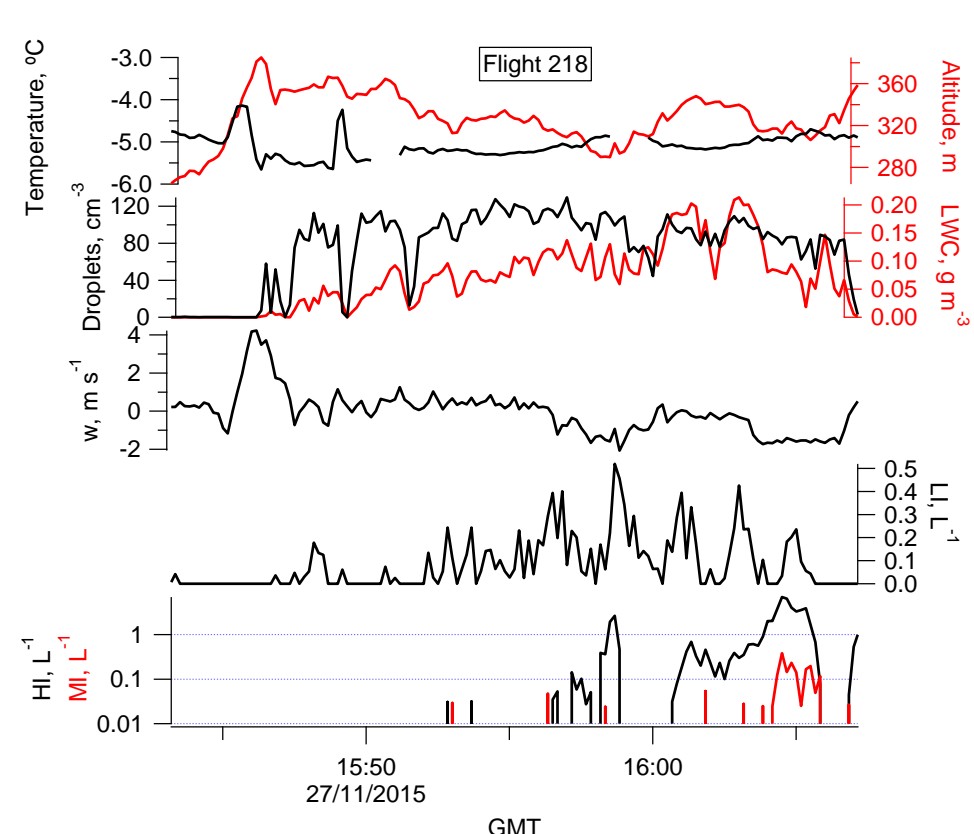

*Figure 8. Time series of microphysical parameters during a constant altitude run at -5°C*

*(400 m) during flight 218.*

During MAC there was a trend towards higher ice concentrations in both updrafts and
downdrafts compared to quiescent regions of the clouds (see Fig. 9 for measurements during
constant altitude runs). Previous measurements have observed secondary ice production in
convective regions of mid-latitude stratus (Crosier et al., 2011). The run during Flight 218 at -
5 °C (see Fig. 8) is an example of this where the two peaks at 3:58 (2DS HI maximum = 3 $L^{-1}$)
) and 4:04 (2DS HI maximum = 7 $L^{-1}$) in ice concentration occur in downdrafts of
approximately 1 m $s^{-1}$. In contrast a similar run during Flight 219 (Fig. 7b) showed glaciated
regions not to be associated with vertical motion.



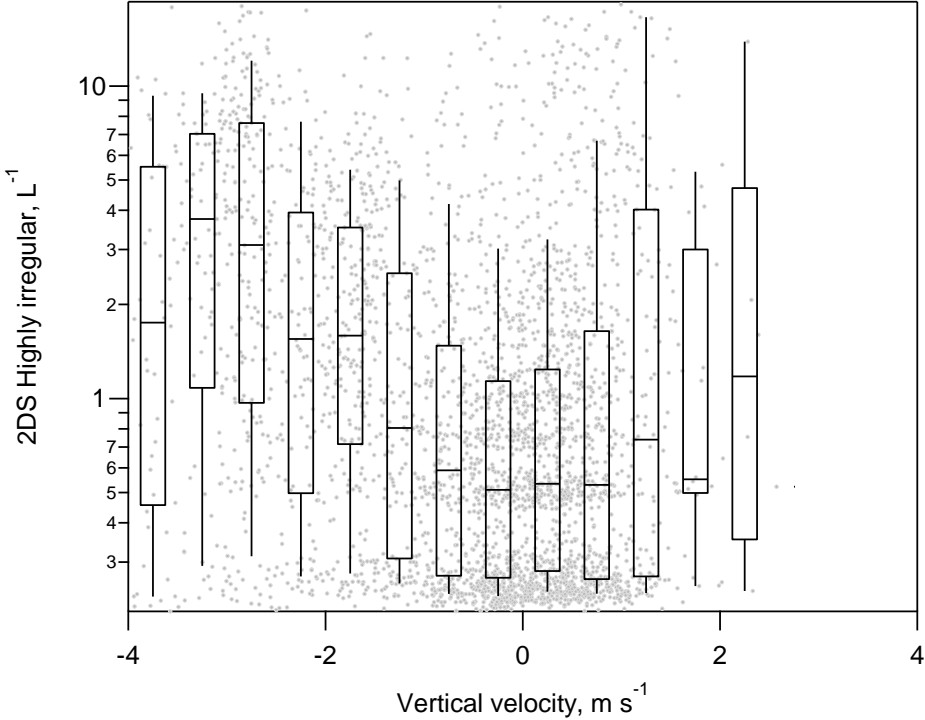

*Figure 9. Box and whisker plots summarising the 1 Hz concentration of highly irregular*

*particles (2DS HI) as a function of vertical velocity. Higher concentrations are observed in*

*updrafts/downdrafts compared to quiescent regions.*

Inspection of the cloud particle images shows that at temperatures higher than -10 °C
columnar crystals appear as the dominant ice crystal habit, with irregular rimed crystals also
widespread. This is illustrated by Fig. 10a showing example images from Flight 218 at -5 °C.
Measurements in Arctic clouds at similar temperatures show that they are similarly dominated
by columnar crystals (Lloyd et al., 2015a). Figure 10b. shows images at -15 °C collected in a
single layer cloud over the Antarctic continent, approximately 300 km south of Halley (Flight
233). This cloud had some columns/needles, but also a high proportion of plates and stellar
crystals. At the lowest sampled temperatures of – 20 °C (Fig. 10c, Flight 226) the ice mostly
consists of rosettes and irregular crystals, which may be aggregates. However, measurements



at these low temperatures were relatively infrequent, and the ice may have been nucleated at
lower temperatures higher in the cloud.

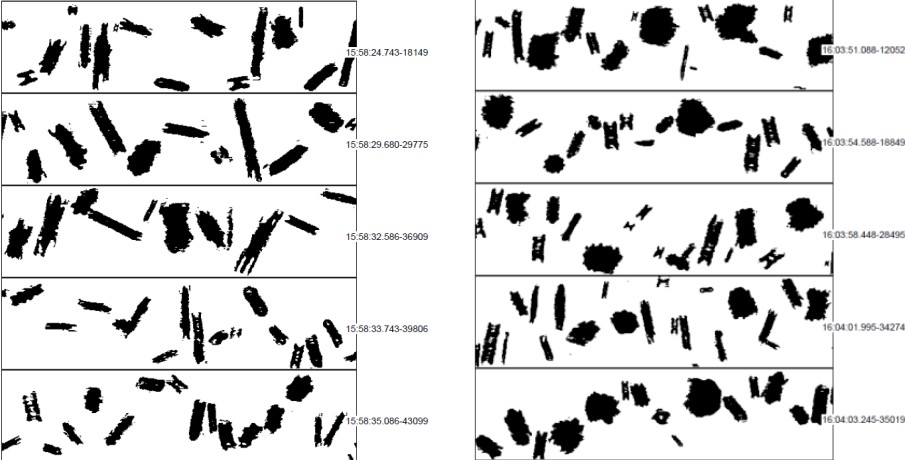

*Figure 10a. 2DS Images of highly irregular particles during a constant altitude run at -5°C*
*(400 m) during flight 218.*

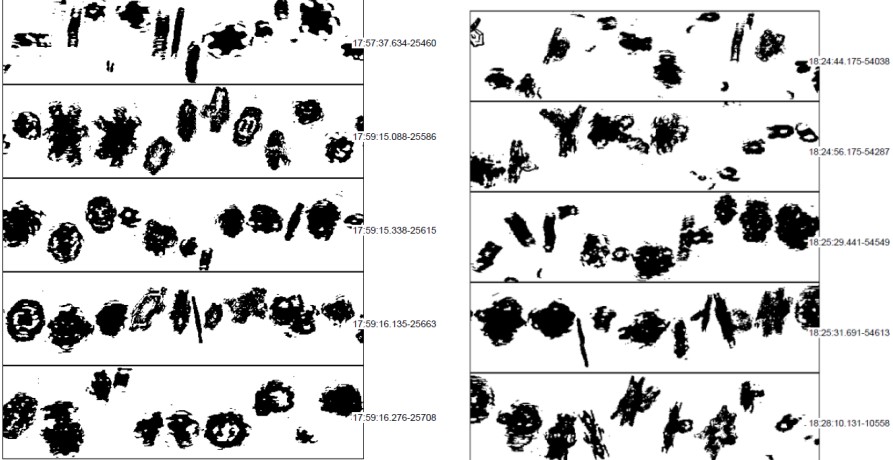

*Figure 10b. 2DS Images of highly irregular particles during a constant altitude run at -15°C*
*during flight 233.*





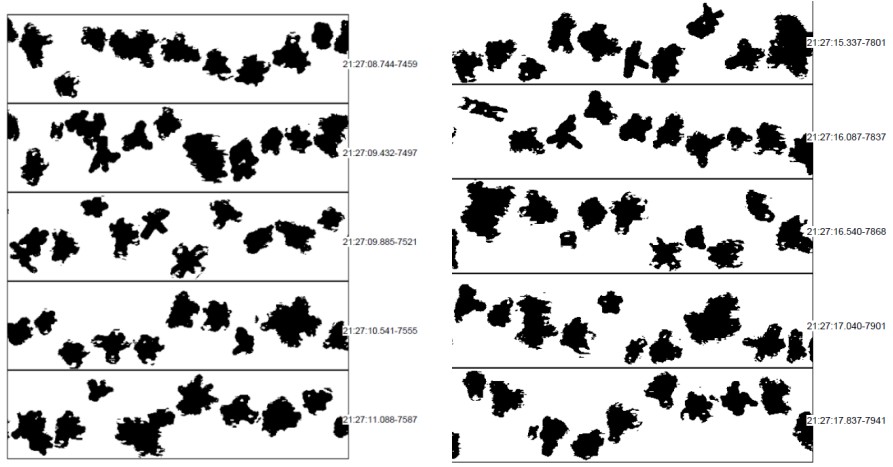

*Figure 10c. 2DS Images of highly irregular particles during a constant altitude run at -20°C*
*during flight 226.*

## 3.2 Aerosol

Vertical profiles of the out-of-cloud aerosol measurements made by the aircraft are shown in
Fig. 11. Out-of-cloud measurements were selected as periods when the LWC was less than
0.001 g m$^{-3}$ and when the 2DS was not detecting particles. Contributions from large, swollen
aerosol particles were also removed when the relative humidity was higher than 90%.
Figure 11a shows aerosol concentrations over the size range from 0.5 to 1.5 μm as observed
by the CAS and GRIMM probes. This size range of aerosols has been shown to best represent
the concentration of INPs in many locations around the world (DeMott et al., 2010).
Concentrations within this size range decrease significantly with increasing height, as would
be expected, through sea spray aerosol being rapidly removed by cloud processing or
sedimentation. Total aerosol concentrations, measured by the CPC, had a median value for the
campaign of 408 scm$^{-3}$ and an inter-quartile range of 260 scm$^{-3}$.
Previous, multi-year measurements of aerosol at the Neumayer coastal Antarctic research
station had a median concentration of 258 cm$^{-3}$. Minimum values (less than 100 cm$^{-3}$) were
typically observed in June/July, while concentrations increased in the austral summer to a
maximum of approximately 1000 cm$^{-3}$ in March (Weller et al., 2011). In winter, aerosol
number and mass were both dominated by sea salt particles (87% by mass, Weller et al.,



2008). Although aerosol composition in summer is more variable, sea salt still accounts for a
significant fraction (50% by mass) but now with a large contribution from non sea salt
sulphate (27% by mass, Weller et al., 2008). Measurements at the coastal Antarctic station
McMurdo show the persistent presence of sulphate aerosol throughout the year (Giordano et
al., 2017). In the winter these particles are highly aged. Sulphate aerosol then increases
through the austral spring/summer, due to enhanced emissions of dimethyl sulphide (DMS)
and methanesulfonic acid (MSA) from phytoplankton in the Southern Ocean (Gibson et al.,
1990; Giordano et al., 2017). Giordano et al. (2017) also report the presence of a sub-250 nm
aerosol population of unknown composition during the winter to summer transition. In
addition a study has observed a significant fraction of organic carbon (>10%) and lower
contributions from sea salt (<10%) in summer marine Antarctic aerosol (Virkkula and Teinil,

12   2006).

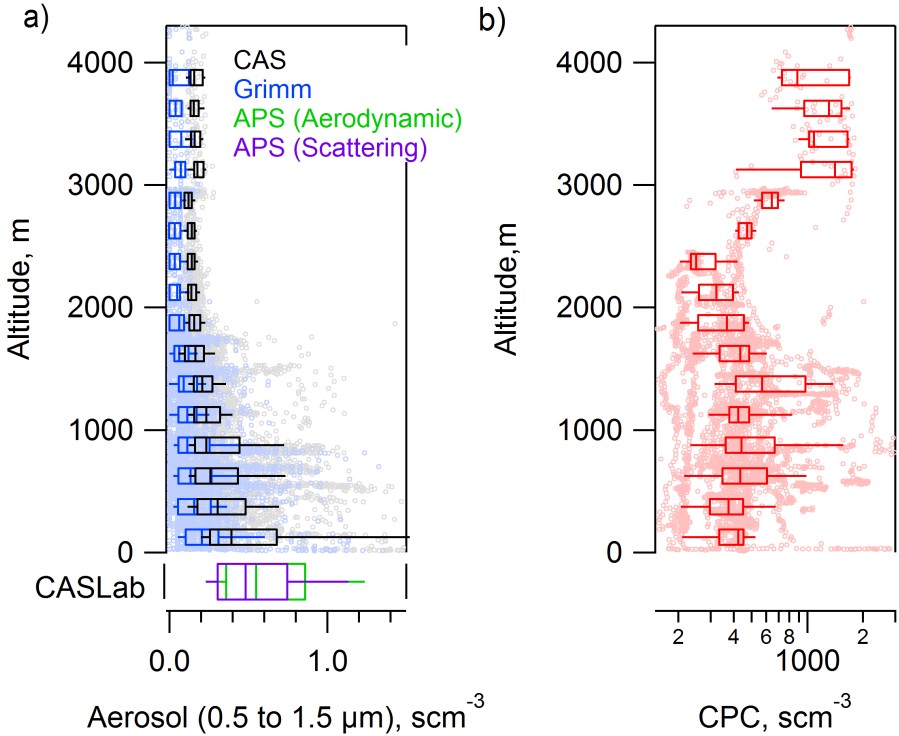

*Figure 11. Aircraft clear sky aerosol concentrations (scm⁻³) altitude profiles. Data are from:*
*a) CAS and GRIMM instruments. Surface concentrations from CASLab are shown for*


*comparison, from the APS; Green - aerodynamic particle size concentrations; Purple –*
*scattering cross section derived particle size concentration measurements; b) Total fine*
*aerosol concentration profiles, from CPC, (D>10 nm).*
During MAC episodic periods were observed with total aerosol concentrations in excess of
1000 scm$^{-3}$. These were often observed above cloud layers. The flights were designed to focus
on cloud regions so may not represent a truly unbiased sample of the atmosphere, but the
results do suggest a link between the observations of high aerosol concentrations and the
presence of clouds. The limited spatial coverage of the aircraft measurements makes
quantifying the extent of these layers uncertain, however they appear to extend over a few
tens of kilometres to a hundred kilometres. At least two instances (flights 218, 219, see Fig.
12) suggest a large layer extending beyond the cloud edge, pointing at the possibility of layers
independent from clouds. The peak concentration usually occurred in the region up to 200 m
above the cloud top (e.g. Flight 219). Some layers showed a clear drop in relative humidity
(e.g. from 90% to 30%, e.g. during flight 220, 221, and 222) generally related to a clear
temperature inversion, while other layers showed a much smaller decrease (by 10%) in
relative humidity compared to the cloud underneath (e.g. flight 217, 218, 219). No clear
systematic relationship was observed with respect to the vertical wind velocity (turbulence).
The role of these particles as CCN/INPs is currently uncertain due to the lack of information
about their composition.



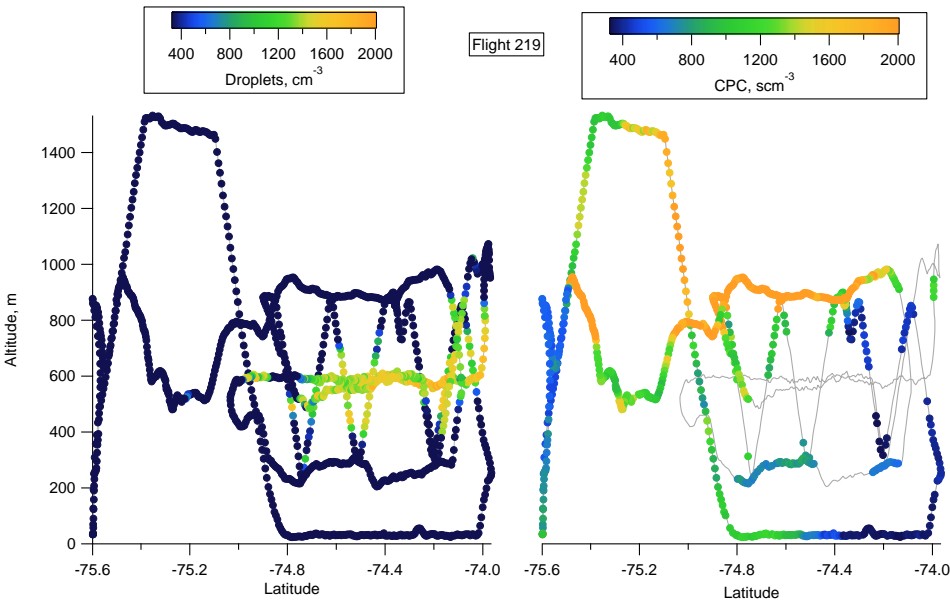

*Figure 12. Latitudinal cross-sections of Flight 219 coloured by droplet concentration (left*
*panel) and total aerosol concentrations out of cloud (right panel). Grey lines shows the flight*
*track. These show a layer of high aerosol concentrations above the cloud top.*
Average total concentrations of UV-fluorescent aerosols (measured at CASLab with the
WIBS) over the campaign period were ~ 1 $L^{-1}$, which was < 2% of the total particle
concentration. Of these 0.01 $L^{-1}$ were identified as likely primary biological aerosols using the
analysis described by Crawford et al. (2015). During some Easterly and Westerly wind
events, however, enhanced concentrations of the order of 5±7 $L^{-1}$ could be observed.
**3.3    Cloud condensation nuclei (CCN)**
Figure 13 (bottom panel) summarises the CCN measurements at the CASLab. The bottom
panel shows the CCN at 5 different super saturations (0.05%, 0.13%, 0.20%, 0.26% and
0.34%). The hygroscopicity parameter κ is used to examine the effect chemical composition
has on the CCN activity of aerosol particles. The derived κ values represent the average
hygroscopicity of the volume-weighted fractions of the individual aerosol components. Non-
hygroscopic components have a κ value of 0.  Highly CCN active salts have κ values between





0.5 and 1.4, sodium chloride (NaCl) has a κ of 1.28 (measurement range 0.91 to 1.33).
Organic species have values generally between 0.01 and 0.5 (Petters and Kreidenweis, 2007).
The median κ value during MAC was 0.64 (inter-quartile range = 0.34, mean = 0.69),
suggesting that this location is dominated by hygroscopic components, such as sea-salt and
sulphate. Andreae and Rosenfeld (2008) review CCN measurements and find that κ values
from marine locations generally cover a relatively narrow range of 0.7 ± 0.2, compared to 0.3
± 0.1 for continental aerosols. A global model study subsequently presented a mean κ value of
0.92 (0.09 at 1σ) at the surface and 0.80 (0.17 at 1σ) within the boundary layer over the
Southern Ocean (Pringle et al., 2010), only marginally higher than our MAC observations.

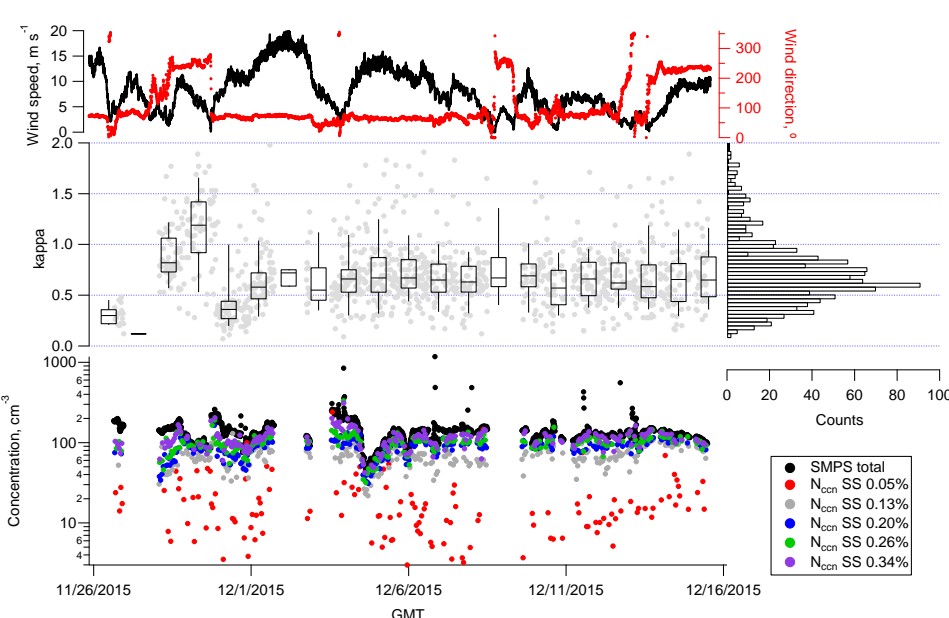

*Figure 13. The top panel shows the time series of wind speed (black line) and direction (red*
*markers) at the CASLab. The middle panel shows the time series of the hygroscopicity*
*parameter κ. The box and whisker plots summarise the variability in κ for each day, while the*
*right panel shows a histogram of κ for the whole measurement period. The bottom panel*
*shows the total aerosol number from the integrated SMPS measurements (30 to 500 nm, black*
*dots) and the CCN concentrations at 5 different supersaturations (SS, coloured dots from 0.05*
*to 0.34%).*

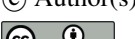



As shown in Fig. 13 there was a period of increased hygroscopicity on 28 and 29 November
2015, with a median κ of 1.18 on 29 November. During this period there was a westerly wind.
This changed to an easterly on 30 November 2015, which coincided with a decrease in
hygroscopicity to a median κ for the 30 November of 0.36. Between the approximate
headings 210° to 25° the CASLab lies between 30 and 60 km from the Weddell Sea. In
contrast, within the sector 30° to 60° it lies several hundred km across the Brunt Ice Shelf
from the Weddell Sea. To the south east of the CASLab lies the Antarctic Continent.
However after 30 November 2015 the hygroscopicity was relatively consistent and does not
show a significant relationship with the wind direction. For example, on the 14 and 15
December 2015 there was a westerly wind but the median κ for these days of 0.66 and 0.65,
respectively, was similar to the campaign median (0.64).
**3.4    Ice nucleating particles (INPs)**
Ice nucleating particles (INPs) could not be directly measured on the aircraft during MAC.
Instead we compare the cloud ice crystal concentrations with two parameterisations that are
commonly used to predict INP concentrations. DeMott et al. (2010) compiled INP
measurements from a range of locations around the world and derived a relationship using
aerosol concentrations (within the size range 0.5 to 1.6 μm) and temperature that could
explain the INP variability within their dataset to better than a factor of 10. For a broad
comparison with the MAC dataset we evaluate DeMott et al. (2010) for a high (1 scm$^{-3}$, dark
blue lines, Fig. 5b) and low (0.1 scm$^{-3}$, light blue lines, Fig. 5b) aerosol case. Cooper (1986)
describes a simple INP parameterisation using only the ambient temperature, which is often
used in the Weather Research Forecasting model (WRF) (Morrison et al., 2009). The
concentration of INPs from Cooper (1986) is shown as a red line in Fig. 5b. It should be noted
that neither of these parameterisations use Antarctic measurements. Given the marine location
of the flights it is likely that these parameterisations may represent overestimates of the true
INP concentration, since the number of INP in sea spray aerosol is generally several orders of
magnitude lower than the number of INP in aerosol in the continental boundary layer (DeMott
et al., 2015). The DeMott et al. (2010) parameterisation was derived using measurements at
temperatures lower than -9°C, while Cooper (1986) used measurements below -5°C. For
comparison they are extrapolated to higher temperatures and are therefore subject to increased
uncertainty.



As shown in Fig. 5b, given the uncertainty in both parameterisations and the challenges with
making a direct comparison with the measurements it is plausible that the observed ice
concentrations at temperatures lower than ca -10 °C could be explained by primary ice
production. However above this temperature the measured ice concentrations diverge from
the predicted INP by 1 to 3 orders of magnitude, suggesting that secondary ice production is
becoming increasingly dominant.
Below -9 °C, where secondary ice production is likely to be less significant, Listowski and
Lachlan-Cope (2017) found that the number of INP predicted by DeMott et al. (2010) gave
better agreement with observed ice concentrations over the Antarctic Peninsula compared to
INP parameterisations that only use the ambient temperature as input. For MAC, each in
cloud data point was compared with the closest (in time) out-of-cloud aerosol measurement (1
minute average, RH < 90%). Data points were excluded from the comparison if no out-of-
cloud aerosol measurements were made within 10 minutes of the in-cloud measurement. No
clear relationship was found between the local aerosol concentrations and the ice
concentrations ($R^2$=0.02 for the above cloud aerosol in the size range 0.5 to 1.6 μm). During
MAC, the majority of cloud measurements showed no ice (see Fig. 3) suggesting that the
Antarctic is a very low INP environment. As a result, all conventional INP schemes will
likely overestimate the true concentrations.
**3.5   Airmass history**
To examine how aerosol and cloud properties vary with airmass history we perform back
trajectory analysis using the UK Met. Office's NAME model (Numerical Atmospheric
Dispersion Modelling Environment) (Jones et al., 2007) using Met Office Unified Model
(UM) meteorological fields. Five-day retroplumes were determined by releasing 10000
particles in the model at locations coincident with the aircraft's position. Here we examine the
relative sensitivity to surface emissions from the following regions; the Antarctic continent,
sea ice, Southern Ocean, ice-shelf and South America. The numbers of particles near the
surface (0 to 100 m) over each geographic region was summed every 15 minutes as the
particles were dispersed five-days backwards in time. For each region, the time integration of
particles over the region was divided by the total number of particles appearing in the whole
domain to determine fractional contributions (see Fleming et al., 2012). Shape files





representing the monthly averaged sea ice extent from Polarview and geographical contour
files for the Antarctic plateau, the permanent sea ice (ice shelves and permanent sea ice) and
the American continent were used to determine the passageway of the air masses at surface
levels sampled by the aircraft. This analysis was repeated for particles released at 60s
intervals along the flight track to determine a time series of contributions from each
geographic region.
Figure 14 shows vertical profiles of the aerosol from the CAS (0.5 to 1.5 µm, relative
humidity < 90%) when there was high (>50%, red markers) and low (<50%, blue markers)
surface influence from the Southern Ocean, the sea ice and the Antarctic Continent. There is a
broad trend of higher aerosol concentrations over this size range with greater contributions
from the Ocean and sea ice, indicating significant emissions of sea salt/sulphate aerosol.
Concentrations decrease with increased contributions from the continent, indicating a lack of
sources in this region. These relationships are more distinct when the aircraft was sampling at
low altitude, above approximately 1000 m the concentrations are less dependent on airmass
origin due to their lower surface influence. This analysis was repeated using total aerosol
concentrations from the CPC (Fig. 14). Similar to the CAS, higher concentrations were
observed when there was greater influence from the Southern Ocean, with the differences
again most distinct for the low altitude measurements. However, CPC concentrations are
found to be less dependent on the influence of the sea ice and the Antarctic Continent.





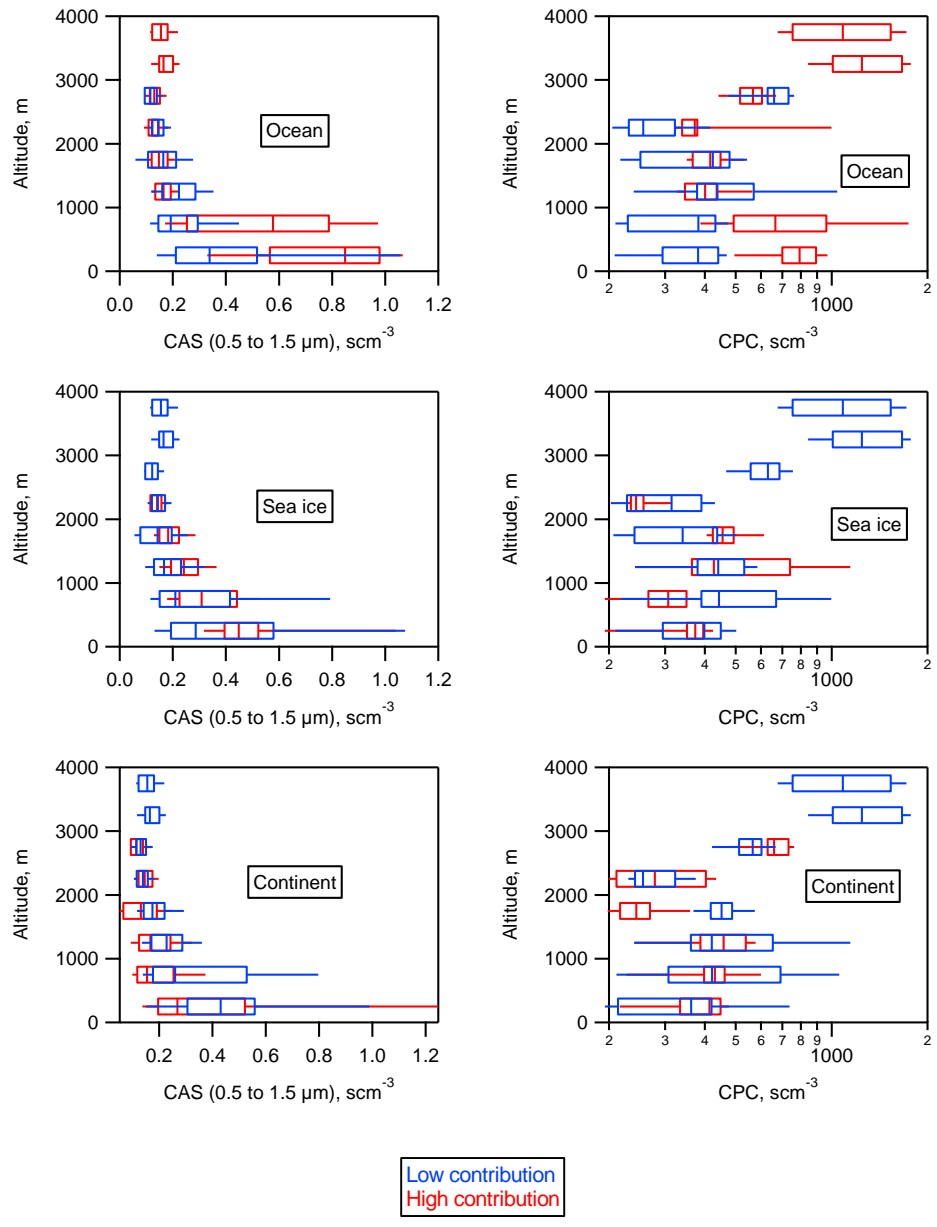

*Figure 14. Altitude profiles of CAS aerosol over the size range 0.5 to 1.5 µm (left panels) and*

*total aerosol, greater than 10nm from the CPC (right panels). The measurements have been*

*partitioned into periods when the airmass had a high (red) and low (blue) contributions from*

*different geographic regions (see text for details).*



Compared to the aerosol measurements the concentrations of cloud droplets and 2DS irregular
particles are found to be less dependent on airmass history. Figure 15 shows these variables as
a function of the relative surface influence from the Southern Ocean, sea ice and the
continent. The concentration of ice in the clouds is found to decrease for airmasses with
increasing influence from the ocean. However, due to ice in the clouds being relatively
infrequently observed the significance of this relationship cannot be determined. The effects
of airmass history cannot easily be deconvolved from differences in sampling strategy or
cloud properties (e.g. humidity, temperature, dynamics, and secondary ice production). Most
of the flights were conducted over sea ice, meaning that near field influences may be
obscuring any relationship with airmass origin.





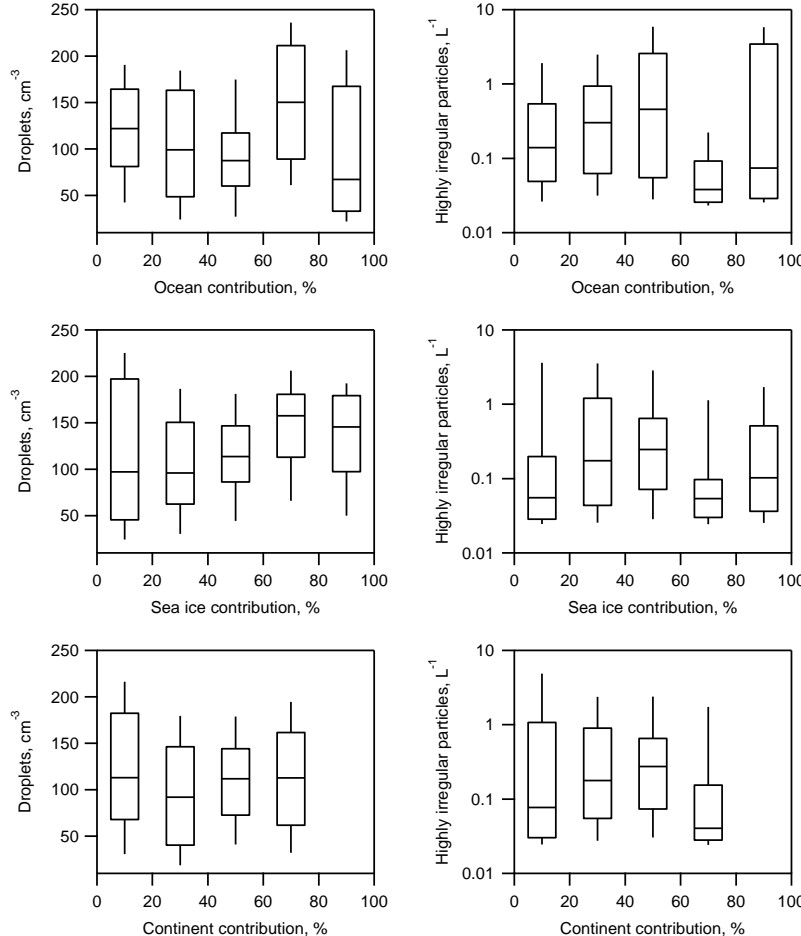

*Figure 15. The concentration of cloud droplets and 2DS highly irregular particles as function*
*of the airmass's contribution from the Southern Ocean, sea ice and the continent (see text for*
*details). Boxes give the 25th and 75th and the whiskers are the 10th and 90th percentiles for*
*each regional contribution bin.*
**4    Discussion**
Ice in the clouds exhibited a high degree of variability, occurring in small patches. Constant
altitude runs by the aircraft through clouds at slightly supercooled temperatures (> -10°C)
showed ice-free regions with patches of high ice concentrations (>1 L$^{-1}$). This variability is
shown to exist over small spatial scales and may be a consequence of very low INP



concentrations, where secondary processes may significantly amplify small differences in INP
concentrations. This makes predicting in detail where ice will form in a given cloud extremely
challenging. A detailed understanding of where the first ice will occur and also the conditions
required for secondary production is needed. Here we examine this variability and discuss
some of the potential controlling factors.
**4.1  First Ice**
First we examine the nature and sources of the INP. Global primary ice nucleation below
approximately -15°C is thought to be dominated by soot and mineral dusts (Möhler et al.,
2006; Murray et al., 2012; Niemand et al., 2012). However, this is colder than the cloud top
temperatures generally observed during MAC. Biological species (pollen, bacteria, fungal
spores and plankton) are the only INP that are known to be active at temperatures higher than
approximately -15°C (Alpert et al., 2011; Murray et al., 2012; Wilson et al., 2015). Bioaerosol
measurements at the CASLab show episodic high concentrations up to several per litre. This
temporal variability in bioaerosol may be analogous to the spatial variability of the ice
crystals observed in the clouds.  Source apportionment of the bioaerosol at Halley is uncertain
with the available dataset, but may include contributions from 1) the re-suspension of material
from the local ice and snow surface, 2) coastal ice margin zones in Halley Bay where bird
colonies are present and 3) long-range transport. The bioaerosol measurements will be
presented and discussed in detail in a separate paper.
It is possible that the cloud layers sampled in MAC are seeded by precipitation from higher
layers where the temperatures are low enough for dust to be active as an INP. During MAC
the flights were designed so that measurements were performed between cloud layers to
determine whether ice seeding from the upper layers was occurring. The frontal cloud
sampled in flight 224 showed extensive ice precipitating between cloud layers and the cloud
top temperature (below -20 °C) was sufficiently low for dust to be a potential source of ice
nuclei. In the case of stratus clouds, those were not found to be seeded by layers at low
enough temperatures for any dust to be active as an INP. Furthermore, single layer clouds
such as those sampled in flights 219 and 227 still showed the patchy ice behaviour.
Detailed measurements of aerosol composition were not available on the aircraft. No clear
relationship could be identified between the local aerosol concentrations and the presence of





ice in the clouds. However, only a small proportion of the total aerosol population are
expected to be INP. Below ca 2000 m (where most of MAC measurements were performed)
there is a broad trend of ice being more frequent with decreasing altitude. A similar
relationship is observed for the concentration of particles between 0.5 and 1.6 μm (Fig. 4).
However, this may in part be due to secondary ice production being efficient at these
relatively high temperatures. Jackson et al. (2012) found a correlation (R=0.69) between the
above cloud aerosol (0.1 < D < 3 µm) and ice concentrations in Arctic stratocumulus clouds.
However these clouds were generally at lower temperatures (cloud top temperature < -10°C)
than those during MAC and as a result are likely to have a higher proportion of primary ice
production.
The surface may also be an ice crystal source either through blowing snow (Ardon-Dryer et
al., 2011) or frost flowers (Gallet et al., 2014; Lloyd et al., 2015b). These will be most
important for clouds in contact with the surface (Vali et al., 2012), but may also be relevant
for low clouds when the humidity is sufficiently high that the crystals do not evaporate whilst
being transported to the cloud base (Geerts et al., 2015). Space-borne lidar measurements of
blowing snow over Antarctica found the thickness of these layers ranging between their
detection limit (30 m) up to 1000 m, with an average thickness of 100 m. Approximately 71%
of these layers were less than 100 m thick and 25% were between 100 and 300 m thick (Palm
et al., 2011). Similarly, lidar measurements at the South Pole found that layers were generally
less than 400 m thick (63%), but could be up to 1000 m thick. Blowing snow is almost always
constrained to the planetary boundary layer (Mahesh, 2003). The lofting of snow is complex;
it is dependent on a range of variables, including: the snow type and surface meteorology (e.g.
wind speed, turbulent mixing, temperature and humidity). A threshold wind speed of 7 to 10
m s$^{-1}$ is typically required (Dery and Yau, 1999). However, smaller crystals may show
substantial fluxes at lower wind speeds. Aerosol fluxes from evaporated frost flowers have
been estimated at $10^{-6}$ m$^{-2}$ s$^{-1}$ at wind speeds as low as 1 m s$^{-1}$ (Xu et al., 2013).
Evaluating the impact of these mechanisms during MAC is challenging since most of the in-
cloud sampling was performed over snow covered sea ice, making it difficult to attribute local
differences in the microphysics to the surface type. Flight 218 (Fig. 8) is one case where the
first ice development may be due to surface ice crystals. During this flight ice was observed
precipitating below cloud base. The majority of this ice precipitation was detected when
flying over snow covered sea ice rather than open water. This was identified from the



aircraft's forward facing camera and inspection of the surface albedo. Given the relatively low
cloud base (300m), strong surface horizontal winds (5 to 10 m s$^{-1}$) and a relative humidity
approaching 100% it is plausible that ice from the surface (e.g. from blowing snow) could
mix up to cloud base, thus providing the first ice to the cloud. The sublimation rate of an ice
crystal is largely dependent on the humidity. A 100 μm ice crystal at 0°C will have a lifetime
of the order 100s at a relative humidity of 80%. At relative humilities of 90% and 95% the
lifetime can be over 200 s and 400 s, respectively (Thorpe and Mason, 1966). The ice crystals
below cloud had similar habits to those observed in the cloud (a mixture of columns and
rimed crystals) indicating they had not originated from the surface. However, only low
concentrations of primary ice from the surface is needed if the ice is then able to multiply
within the cloud due to secondary processes.
**4.2 Secondary Ice**
Previous ice crystal observations over the Antarctic Peninsula show a similar behaviour to
those during MAC with a peak in ice concentrations (> 1 L$^{-1}$) at approximately -5°C.
Grosvenor et al. (2012) and Lachlan-Cope et al. (2016) attribute this to secondary ice
production through the Hallett-Mossop process, where ice splinters are produced when a
droplet freezes subsequent to colliding with an ice crystal (riming) (Hallett and Mossop,
1974). This can lead to rapid ice multiplication as the splinters freeze further drops, resulting
in more splinters. Laboratory experiments suggest that this process is efficient over a narrow
temperature range (-8 to -3 °C) with a peak at -5 °C (Mossop, 1976). Images from the 2DS
probe at temperatures higher than -10°C generally show rimed crystals and small columns
(Fig. 10a). These habits are generally observed when the Hallett-Mossop production
mechanism is thought to be occurring (Crosier et al., 2011; Lloyd et al., 2015a).
A number of other secondary ice mechanisms have previously been identified, these include:
large drops producing ice splinters when they freeze (Lawson et al., 2015); and the break-up
of ice crystals, generally either fragile dendrites due to sublimation, turbulence (Bacon et al.,
1998) or because of collisions between crystals (Yano and Phillips, 2011). However, all these
processes have only been observed to be efficient at temperatures lower than approximately -
10 °C, which is lower than the temperature of the majority of clouds sampled during MAC.
Taylor et al. (2015) suggest that the drop-freezing secondary ice production, identified by





Lawson et al. (2015), may have occurred at temperatures higher than -10 °C in their
measurements of cumulus clouds. However, they were not able to deconvolve its effects from
the Hallett-Mossop mechanism. We have not performed automatic habit recognition on the
2DS images taken during MAC, however, inspecting the images "by-eye" suggests that the
drop shattering events observed by Lawson et al. (2015) were not common during MAC.
The exact requirements for secondary ice production through Hallett-Mossop are still
uncertain. It is thought that only a small of amount of primary ice is needed for it to be
initiated, and recent model studies suggest this could be as low as 0.01 $L^{-1}$ (Crawford et al.,
2012; Huang et al., 2017). Laboratory experiments suggest that production rates are
proportional to the accumulation of large drops (>24 μm) (Mossop and Hallett, 1974).
However, more recent field measurements found that estimated crystal production rates gave
better agreement with observed ice concentrations if this constraint on drop diameter was
removed (Crosier et al., 2011).  Observations of Arctic mixed phase clouds found that the
presence of precipitating ice particles (> 400 μm) was correlated with the number of large
drops (>30 μm), however the precise nucleation mechanism through which this occurred was
uncertain (Lance et al., 2011). During MAC both the analysis of individual case studies and
the statistics for the whole campaign do not suggest that the concentration of large drops and
ice crystals were related.  However, any simple relationship is likely to be complicated as ice
crystal growth will deplete the drops through riming and the Wegener-Bergeron-Findeisen
process. This is shown in Fig. 6 and 7b where the highest ice concentrations correspond to
relatively low droplet concentrations.
Flights 226, 227 and 228 involved sequential vertical profiles to examine the dependency of
ice on the clouds vertical structure. No link was identified between the presence of ice in the
vertical profile and local variations in cloud top temperature. However, since the first ice
occurs over small spatial scales, any relationship may be obscured by the aircraft's horizontal
motion whilst changing altitude. As a result the precise cloud top temperature, and its
variability, directly above the glaciated regions of the clouds is not known.
Higher ice concentrations were observed in updrafts/downdrafts compared to quiescent
regions of the clouds. There are several possible explanations for this; first the more turbulent
conditions may make more primary ice available through greater entrainment of aerosol and
hence potentially more INP into the cloud. Second convective regions may indicate thicker
regions of the cloud and lower cloud top temperature. This may lead to increased primary ice



nucleation as the lower temperatures activate more INPs and the development of larger liquid
droplets. Third, the more turbulent conditions could lead to more efficient ice production due
to ice being rapidly mixed to the Hallett-Mossop zone where concentrations would multiply.
Finally, the riming rate may increase due to a greater number of ice-liquid collisions. More
turbulent conditions may also indicate higher rimer velocity, however laboratory experiments
suggest there is no lower cut-off rimer velocity for Hallett-Mossop to be active (Mossop,

7 1985).

**5    Conclusions**
We have reported observations of cloud and aerosol properties over coastal Antarctica and the
Weddell Sea. The aerosol was predominantly hygroscopic in nature, with κ being consistent
with previous measurements and model predictions for remote locations dominated by marine
emissions. The concentration of large aerosols (0.5 to 1.6 μm) decreased with altitude, as
would be expected, through sea salt/sulphate aerosol being rapidly removed by cloud
processing or sedimentation. Higher aerosol concentrations were observed in airmasses that
travelled over the Southern Ocean/sea ice compared to those from the main Antarctic
Continent.
In contrast to the aerosol concentrations, the droplet and ice concentrations showed minimal
dependence on airmass origin. The cloud types were generally stratus, both single and
multiple layers, at temperatures between -20 and -3 °C. These were dominated by super-
cooled liquid drops, with a median concentration of 113 cm$^{-3}$. Droplet concentrations were
relatively consistent throughout the campaign with an inter-quartile range of 86 cm$^{-3}$. The
exceptions to this were cases when the concentrations became depleted by high ice
concentrations, and also during Flight 217 when anomalously high droplet concentrations
were observed; this was associated with an enhanced aerosol layer below the cloud layer.
Ice in the clouds exhibited a high degree of inhomogeneity occurring in small patches. Below
ca 2000 m ice was more frequent at higher temperatures, however even within the -8 to -3 °C
temperature range where Hallett-Mossop secondary production is most active, the clouds
were predominantly liquid. When ice was present within the temperature range -8 to -3 °C it
seems likely that secondary ice production, through the Hallett-Mossop process, resulted in
concentrations that were 1 to 3 orders of magnitude higher than the number of INP predicted
by conventional primary ice nucleation schemes. The source of first ice in the clouds is



currently uncertain. First ice in the clouds often occurs at temperatures above -10 °C, this may
be due to the presence of biogenic particles that are active INP at these temperatures or
alternatively (or indeed simultaneously) ice from the surface (e.g. blowing snow or frost
flowers) could be lofted into the clouds. The drivers of the ice crystal variability were
investigated. No dependence on the droplet spectrum was found. However, higher ice
concentrations were found in updrafts and downdrafts compared to quiescent zones, and
therefore intermittent convective activity may explain the intermittent glaciation of clouds.
This paper has presented the most detailed in situ observations of coastal Antarctic clouds and
their surrounding aerosol properties to date. Upcoming studies will use the MAC observations
to test and improve the representation of Antarctic clouds in numerical weather/climate
models.
**Acknowledgements**
The authors would like to thank Vicky Auld and all the BAS staff who helped in the
Antarctic. The MAC project was funded by the UK Natural Environment Research Council
(Grant number: NE/K01482X/1).





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
