# Peer review of "In situ measurements of cloud microphysics and aerosol over"

_Atmospheric Chemistry and Physics, 2017_

## Referee Comment (RC1) · Anonymous Referee #1 · 9 May 2017

Overview

This manuscript presents an analysis of cloud and aerosol measurements collected during the Microphysics of Antarctic Clouds field campaign. The main focus of the analysis presented is on extensive airborne observations from 24 flights, primarily in conditions containing stratiform cloud layers, and these are supplemented by measurements made at a surface site. The observations show that the clouds are dominated by liquid water with variable (and typically low) concentrations of ice particles, suggesting that there are limited sources of primary ice nuclei that are active in the temperature range of these clouds. The ice particle concentrations tended to increase in the H-M temperature zone, suggesting that secondary ice production can play a role in these

clouds.

The main strength of the paper lies in the fact that there is a scarcity of in-situ observations of aerosol and clouds in the Antarctic, resulting in a very limited number of observational constraints that can be used to evaluate NWP and climate models in this region. The novel observations in this paper certainly have the potential to be useful for model evaluation studies and increase our knowledge of some cloud and aerosol microphysics parameters in the region. I do however think that some additional analysis of the data is required before the manuscript can be published in ACP (see comments below).

Main comments

1. Introduction: The authors give some background information on previous Antarctic INP measurements, but there is an absence of information on previous CCN measurements. The CCN are key to the liquid dominated clouds studied in this paper. The introduction should be expanded to include additional information on past results on Antarctic CCN data to put these new observations into context.

2. Section 2.1: The information on the meteorological and cloud conditions that were present during the observation period needs to be strengthened significantly. I realise that there were a lot of flights and that it is not straightforward to summarise this information in a paper, but the very short bit of text on page 6 (lines 1 to 5) is inadequate. Perhaps including additional information in a supplement would be worthwhile, such as a surface analysis chart, a satellite image and the back-trajectories calculated in section 3.5 for each case.

3. Vertical profiles of thermodynamic and cloud data for each aircraft flight would also be extremely useful to include, which again perhaps could be included in a supplement. This would enable the reader to put the microphysical measurements into better context with the cloud and meteorological conditions for each case. It would also be extremely useful for model evaluation purposes.

4. Data from all flights are composited and summarised as a function of altitude (Figures 4, 11, 14), yet presumably there is significant day-to-day variability in the cloud top and cloud base heights. The main problem with this approach is that it is difficult to disentangle changes in the in-cloud, above cloud and below cloud measurements (e.g. location of ice and aerosol particles) with variability in the vertical location of the clouds. Have the authors considered normalising the data relative to the position in the cloud (at least for single-layer clouds), which would then be more comparable to previous studies e.g. McFarquhar et al. (2007)? This, in combination with vertical profile plots (comment 3) would give the reader a much clearer picture of the cases sampled by the aircraft.

5. Page 9 line 22: The calculation of ice mass fraction does not include all particles. The ice mass is taken from the sum of the 2DS MI and HI categories, which cover particles larger than about 80 microns. The LWC is calculated from the CAS, which covers particles sizes less than about 50 microns. Furthermore, as discussed earlier in the manuscript the MI category may include large drizzle drops. The CAS would also be expected to measure small ice particles. Can the authors give some measure of the uncertainty in the calculated IMF that would result from defining liquid and ice water content in this way?

6. Figure 4: It would be better if the authors showed the IMF data as a box and whisker plot rather than plot a mean profile which is pretty meaningless, especially above about 1 km altitude. At these altitudes the mean profile shows values between 0.2 and 0.8, yet the majority of the data points look to have values close to 0 or 1 i.e. the clouds appear to be dominated by liquid drops or they are almost completely glaciated at these altitudes.

7. All figures where you include box and whisker diagrams: I would encourage the authors to ensure that there are the same number of data points included in each bin, otherwise the plots can be misleading. For example, when referring to Figure 9 on page 18 line 6, the authors state that "there was a trend to higher ice concentrations in

both updrafts and downdrafts". It is unclear to me if the apparent increasing trend in the updrafts is simply the result of poor statistics. There appear to be very few data points with w > 2 ms-1, which are what I think the authors are using to justify the statement. If the bins were adjusted to include the same number of data points then I suspect that the trend may not be evident.

8. Page 13 line 16: Can the authors use the CIP data for the flights where the 2DS was not operating? This would be of interest as it would extend the analysis to colder temperatures, where there is arguably larger differences in the INP parameterizations.

9. Page 13 line 22: Again no clear trend is evident. There are very few data points at temperatures lower than -15 C and there is a lot of scatter in the data.

10. Page 13 line 24: Is the figure showing the histogram missing? This is key to demonstrate that the ice occurs in small patches in the liquid clouds.

11. Page 15: Why do the authors only sub-select data in the H-M temperature range where secondary ice production might be expected to be enhanced? It would be beneficial to also look to see if there was a relationship outside of this temperature zone e.g. to see if there is any evidence of primary ice being formed from the freezing of drizzle drops (e.g. Rango and Hobbs 1991).

12. Page 18 and figure 9: What is the mechanism by which glaciation would occur preferentially in downdrafts?

13. Figure 9: From the figure, it looks like the frequency of downdrafts measured by the aircraft is much larger than the updrafts. This was surprising to me and I would like the authors to confirm that there is no instrumental bias in the vertical wind measurement. Assuming that there is no bias, can the authors explain the higher frequency of downdrafts in the measurements?

14. Page 19: Is the implication that the columns observed at warmer temperatures are generated by secondary ice processes? If so make that clear to the reader and link to

figure 5 perhaps.

15. Figure 11: Can you include CPC data from the ground site? Also, how is number concentration derived from the aerosol scattering cross-section?

16. Page 21 line 13: Presumably any surface generated aerosol is inhibited from being transported above the boundary layer as there are likely to be strong thermodynamic gradients co-incident with the top of single-layer clouds? If the data were normalised and plotted relative to cloud/boundary layer top (see point 4), you may expect to see sharper vertical changes in the profile of aerosol. These would be smeared out in figure 11 as all flight data are simply plotted relative to altitude and the data are therefore averaged over different boundary layer depths. Also, if the boundary layer was not well-mixed, then this could also result in a drop-off in the concentration of surface generated aerosol with altitude. Can you examine the aircraft thermodynamic data to examine if this was important?

17. Page 23 line 8: Is there any evidence that the enhanced aerosol concentrations observed above cloud are being entrained into the cloud top? The suggested linkage between high aerosol concentrations above cloud top and the presence of clouds is rather tenuous.

18. Page 24 lines 6 to 10: This short paragraph on the WIBS data (which is not very informative) should either be expanded to link to the aircraft measurements or other ground based data, or removed given that a subsequent paper on this data is planned.

19. Page 26 line 1: Can you use back-trajectories to see if there is a different source region for the case where the aerosol hygroscopicity increased between 28 and 29 Nov?

20. Page 27 line 15: What about using the below-cloud aerosol concentration? This is more likely to be relevant, especially for cases were no elevated aerosol layers were observed to be in contact with cloud top.

21. Figure 14: Is the data in the top right panel suggesting that the source of the elevated aerosol concentrations are from the southern Ocean? Are these above cloud?

22. Figure 15: It is somewhat surprising that given that there is evidence of a correlation between aerosol concentrations and source region (Fig 14) that this is not apparent in cloud drop concentration. I would have expected increases in the aerosol concentrations to result in elevated cloud drop concentrations in these liquid dominated layer clouds.

23. The discussion section focuses solely on ice production in the clouds. Given that these are liquid dominated clouds the authors should also include some discussion on the liquid phase.

24. Page 25 line 27: Can this not be estimated given that the aircraft was doing vertical profiles?

Minor comments and technical corrections

1. Page 1 line 23: Additional clarification on what "key processes" means is required.

2. Page 1 line 27: Clarify that the size quoted is particle diameter.

3. Methods: It would seem more appropriate to introduce the NAME model in the methods section instead of in the results section.

4. Page 6 line 22: I don't think the CAS part of a CAPS probe has anti-shatter tips. Please check. Also, was the CDP fitted with anti-shatter tips?

5. Are the aircraft altitudes relative to mean sea level?

6. Page 6 line 28: Can you put a measure of uncertainty on your IWC estimate that results from assuming the Brown and Francis mass-diameter relation?

7. Page 8 line 5: The MI data is only shown in one small subset of data (Fig 8).

8. Figure 2: Is this a flight average or an in-cloud average PSD? Rather than showing

one PSD, can you give a more quantitative measure of the agreement between instru-
ments over all flights e.g. compare drop number and LWC from CDP/CAS, IWC from
2DS/CIP, aerosol concentration from GRIMM/CAS. It is also worth describing why you
choose certain probes in your analysis e.g. CAS appears to be used preferentially for
LWC over the CDP.

9. Page 9 line 10 to 14: Suggest removing if data is not shown/discussed.

10. Figure 4: What is the point of figure 4b? Is it just to zoom in on the data closer to
the surface? If so, why cut-off the x-axis at 0.3 when there are many data points above
this as shown in figure 4a. Consider removing the lower panel.

11. Figure 5: The continuous colorbar used does not adequately discriminate different
flights. Suggest using an alternative way of plotting each flight or remove the colorbar.

12. Page 12 line 10: The authors state the reason is not clear but then go on to say
that higher aerosol concentrations were observed which could explain the higher drop
numbers. Do you see any clear correlation between sub-cloud aerosol and cloud drop
number across the different flights?

13. Figure 10: The figure caption needs to include a lot more information.

14. Page 18 line 10: Replace 3:58 with 15:58 and 4:04 with 16:04 to be consistent with
figure axis label.

15. Throughout the manuscript the authors use the abbreviation "ca". Why not simply
write "circa"? Or consider replacing with "approximately" or "about", both of which are
used in this context more extensively in the English language.

16. The authors switch between using cm-3 and scm-3 when referring to number
concentrations of cloud and aerosol particles. Cloud drop number concentration data
are shown in cm-3 (Figs 5, 6, 8, 12, 15), whereas aerosol number concentration data
are mostly shown in scm-3 (Figs 11, 12, 14), except for CCN concentrations in Fig 13
which use cm-3. I would suggest being consistent in the use of units throughout and

using cm-3. It is the ambient aerosol number concentration on which the cloud drops form (not the value at STP) that is important for cloud drop number concentration for example.

---

## Referee Comment (RC2) · Anonymous Referee #2 · 18 May 2017

The manuscript summarizes the characteristics of the Microphysics of Antarctic Clouds, or MAC campaign conducted in 2015 over coastal Antarctica and the nearby ocean regions. The measurements, comprising cloud and aerosol retrievals from both ground-based and airborne measurements, provide a compelling source of information for this region, that will most likely be of high interest to the atmospheric modelling community for example.

The Authors go further into analyzing the key features in the cloud and aerosol retrievals and the processes affecting them. Interesting features are revealed about the cloud ice mass fraction, ice particle types and the aerosol, even though conclusive explanations for many of the observed cloud features seem elusive. However, much of

this can be rightfully attributed to the extremely challenging conditions as well as the so far quite low number of observations in this region. The descriptions of the instruments and retrieval techniques, as well as the presentation of data are for the most part adequate and very good.

My main concerns about the manuscript are related to the description of the surrounding conditions during these measurements. I do think the manuscript would greatly benefit from a bit more systematic description of the meteorological conditions as well as the structure of the clouds sampled during the campaign. This could potentially help with the interpretation of the results, many of which are now based simply on microphysical retrievals put together from all available flights.

Below, I will try to summarize these points with more specific comments, followed by minor and technical comments.

1. The manuscript does not provide very detailed information about the height and depth of the sampled cloud layers. If there are considerable differences in the altitude and depth of the cloud layer, this would imply differences in the cloud dynamics as well as in the large-scale meteorological setting from one flight to the next, and would therefore be worth a look to support the subsequent analysis. It is also rather difficult to follow whether the analyzed data (in general but also in the few flight specific examples) represent the cloud base, in-cloud or cloud top conditions. For example, the vertical profile of ice mass fraction given in Fig 4 is interesting information, but it would make it far more interesting, if that data could be put in the context of sampling level with respect to the vertical extent of the sampled cloud.

2. Would it be possible to consider the existence of multiple cloud layers for all of the flights? It is commented in Section 4.1 that seeding effects were not detected apart from the frontal cloud case. However, could the possibly overlapping cloud layers affect the radiation budget of the sampled clouds, that might impact mixing and perhaps entrainment and thus the cloud properties?

3. While the manuscript does consider the impact of airmass history on particle number concentrations in Section 3.5, it does not clearly outline how these shifts affect other aspects, such as large-scale meteorological forcing, which could impact the cloud properties. Generally, I think it would add great value to put these observations into context by including some more detailed information about the meteorological conditions during the campaign.

4. Related to the above, it would be interesting to couple meteorological data (e.g. wind speed) with estimates of the cloud altitude (as per the comments no. 1) in order to estimate the potential of blowing snow to affect the measured cloud properties. This possibility is considered in Section 4.1 but explicitly only for one flight. A more detailed evaluation of the prevailing conditions would allow a broader consideration of at least the possibility of blowing snow contribution also for other flights.

Other minor and technical comments

1. The notation scm-3 is used throughout the paper for the units of aerosol concentration, apparently referring to concentration defined in STP conditions, yet for cloud droplets the more commonplace cm-3 is used. I think the STP-related notation should be clearly defined and it should be made very clear where each of the notations are used to avoid confusion. However, I do think the best option by far would be to just use the same units (cm-3) for all concentrations.

2. Page 7, line 9: Please add a short definition for the circularity.

3. Fig 4, "Radar altitude" vs "altitude", are the panels from different datasets? Please explain.

4. Page 13 lines 24-27: Histogram of the spatial extent of ice: which figure should I be looking at here? If it is missing, please consider adding a figure showing this.

5. Page 15, and fig 6: Why do you limit the analysis to the temperature range -8...-3 in this particular case? Please elaborate.

[Figure]

6. The terminology regarding Fig 7 is confusing: page 16, line 9 "particle size distri-butions", page 17, line 2 "droplet spectrum" and just "size distributions" in the caption. Please try to use more consistent terminology here so it's easier to read and to know exactly what you refer to.

7. Fig 7: You don't mention the dashed lines in the caption or elsewhere in the text. Please add an explanation for these or remove them from the figure.

8. Page 17, line 5: Could you please add some numbers to make a more clear case for the droplet depletion?

9. Page 19: Please consider adding a dedicated subsection for the analysis of the cloud particle images. Coming directly after the quantitative results on ice fractions and number concentrations it feels a little out of place to me.

---

## Author Comment (AC1) · 28 Jul 2017

We would like to thank both referees for their constructive comments about our manuscript. We now address their comments individually. For clarity referee comments are coloured red and responses are in black.

Referee 1

Overview
This manuscript presents an analysis of cloud and aerosol measurements collected during the Microphysics of Antarctic Clouds field campaign. The main focus of the analysis presented is on extensive airborne observations from 24 flights, primarily in conditions containing stratiform cloud layers, and these are supplemented by measurements made at a surface site. The observations show that the clouds are dominated by liquid water with variable (and typically low) concentrations of ice particles, suggesting that there are limited sources of primary ice nuclei that are active in the temperature range of these clouds. The ice particle concentrations tended to increase in the H-M temperature zone, suggesting that secondary ice production can play a role in theseclouds. The main strength of the paper lies in the fact that there is a scarcity of in-situ observations of aerosol and clouds in the Antarctic, resulting in a very limited number of observational constraints that can be used to evaluate NWP and climate models in this region. The novel observations in this paper certainly have the potential to be useful for model evaluation studies and increase our knowledge of some cloud and aerosol microphysics parameters in the region. I do however think that some additional analysis of the data is required before the manuscript can be published in ACP (see comments below).

Main comments
1. Introduction: The authors give some background information on previous Antarctic INP measurements, but there is an absence of information on previous CCN measurements. The CCN are key to the liquid dominated clouds studied in this paper. The introduction should be expanded to include additional information on past results on Antarctic CCN data to put these new observations into context.

The description of previous aerosol measurements in Sect 3.2 has been moved to the introduction and extended to include references discussing aerosol hygroscopicity in the region.

2. Section 2.1: The information on the meteorological and cloud conditions that were present during the observation period needs to be strengthened significantly. I realise that there were a lot of flights and that it is not straightforward to summarise this information in a paper, but the very short bit of text on page 6 (lines 1 to 5) is inadequate. Perhaps including additional information in a supplement would be worthwhile, such as a surface analysis chart, a satellite image and the back-trajectories calculated in section 3.5 for each case.

As suggested we have added surface pressure charts and HYSPLIT back trajectories for each flight in the supplementary material. A table has been added to the manuscript (Table 1) detailing cloud base/top height and temperatures. Also included is information about whether multiple cloud layers were present.

3. Vertical profiles of thermodynamic and cloud data for each aircraft flight would also be extremely useful to include, which again perhaps could be included in a supplement. This would enable the reader to put the microphysical measurements into better context with the

cloud and meteorological conditions for each case. It would also be extremely useful for model evaluation purposes.

The aim of this paper is to provide a statistical overview of all measurements. Showing thermodynamic and cloud data profiles for each individual flight would make the paper overly long and repetitive. The data used in this paper will be made publically available at the Centre for Environmental Analysis archive allowing direct comparisons to be made between measured and modelled microphysics.

4. Data from all flights are composited and summarised as a function of altitude (Figures 4, 11, 14), yet presumably there is significant day-to-day variability in the cloud top and cloud base heights. The main problem with this approach is that it is difficult to disentangle changes in the in-cloud, above cloud and below cloud measurements (e.g. location of ice and aerosol particles) with variability in the vertical location of the clouds. Have the authors considered normalising the data relative to the position in the cloud (at least for single-layer clouds), which would then be more comparable to previous studies e.g. McFarquhar et al. (2007)? This, in combination with vertical profile plots (comment 3) would give the reader a much clearer picture of the cases sampled by the aircraft

For the single layer clouds we have added a plot showing IMF (and liquid water content/effective radius) as a function of normalised position within clouds.  The following text has been added to manuscript:

"Figure 3b, shows ice mass fraction measurements in single layer clouds as a function of the normalised position within the cloud, Zn.

$$Zn = \frac{Z - Z_B}{Z_T - Z_B},$$

Equation 2

where Z is the altitude, $Z_B$ and $Z_T$ are cloud base and cloud top altitude, respectively. We note that there is some uncertainty in determining cloud base/top due variability in the cloud and also incomplete sampling (this uncertainty is estimated in Table 1). The clouds were dominated by liquid drops throughout, while ice was more prevalent lower in the clouds. The relationship between ice mass fraction (IMF) and Zn over the range 0<Zn<1 can be approximated by the equation:

$$IMF = 0.177 + 0.360Zn + 0.244Zn^2$$

Equation 3

This is shown as a red line in Fig. 3b. Figure 3c and d show that both liquid water content and cloud drop effective radius increase closer to cloud top. The effective radius increases from 4 ± 2 µm near cloud base to 8 ± 3 µm near cloud top.

Measurements in Arctic stratus/stratocumulus generally find these clouds to be similarly dominated by liquid drops (McFarquhar and Cober, 2004; McFarquhar et al., 2007; Lloyd et al., 2015a). A polynomial relationship derived during the Mixed-Phase Arctic Cloud Experiment (M-PACE) is also shown as a blue line in Fig. 3b (McFarquhar et al., 2007). McFarquhar et al. (2007) show a trend of increasing IMF with increasing distance from cloud

top (and increasing temperature). Glaciated conditions were observed during 23% of their measurements. This is significantly more than during MAC, possibly due to lower INP concentrations available for primary ice development in the Antarctic compared to the Arctic, but differing sampling strategies may also contribute to this difference. "

[Figure]

*Figure 3a. Ice mass fraction as a function of altitude and b) normalised position within the cloud (Zn). c) and d) show similar plots for liquid water content and effective radius from the CAS probe. Boxes are the 25th and 75th percentiles, the whiskers are the 10th and 90th percentiles.*

5. Page 9 line 22: The calculation of ice mass fraction does not include all particles. The ice mass is taken from the sum of the 2DS MI and HI categories, which cover particles larger than about 80 microns. The LWC is calculated from the CAS, which covers particles sizes less than about 50 microns. Furthermore, as discussed earlier in the manuscript the MI category may include large drizzle drops. The CAS would also be expected to measure small ice particles. Can the authors give some measure of the uncertainty in the calculated IMF that would result from defining liquid and ice water content in this way?

It is not possible to unambiguously determine the phase of particles smaller than about 80 µm. However, under mixed phase conditions ice will rapidly grow at the expense of liquid drops. Given a typical crystal growth rate of 1 µm s$^{-1}$, within a couple of minutes crystals will grow to a size where there phase can be determined by the 2DS. Therefore it is unlikely that the number of crystals smaller than 80 µm is significantly larger than those greater than 80 µm. As a consequence crystals smaller than 80 µm are only expected to make a small contribution to the total ice mass.

As described in the text the concentration of MI particles was generally significantly less than HI particles. The mean ratio HI:MI for the campaign was 7. If we assume that all MI particles are large drizzle droplets (inspection of the MI images suggests this is unlikely) then 5% of measurements are classified as fully glaciated (0.9<IMF<=1) compared to 6% if MI images are classified as ice.

6. Figure 4: It would be better if the authors showed the IMF data as a box and whisker plot rather than plot a mean profile which is pretty meaningless, especially above about 1 km altitude. At these altitudes the mean profile shows values between 0.2 and 0.8, yet the majority of the data points look to have values close to 0 or 1 i.e. the clouds appear to be dominated by liquid drops or they are almost completely glaciated at these altitudes.

As suggested this has been changed to a box and whisker plot.

7. All figures where you include box and whisker diagrams: I would encourage the authors to ensure that there are the same number of data points included in each bin, otherwise the plots can be misleading. For example, when referring to Figure 9 on page 18 line 6, the authors state that "there was a trend to higher ice concentrations in both updrafts and downdrafts". It is unclear to me if the apparent increasing trend in the updrafts is simply the result of poor statistics. There appear to be very few data points with w > 2 ms-1, which are what I think the authors are using to justify the statement. If the bins were adjusted to include the same number of data points then I suspect that the trend may not be evident.

For each box and whisker plot the raw data is also shown so that the reader can see the strength of the relationship. Also note that Figure 9 has been removed from the manuscript (please see our response to comment 12).

8. Page 13 line 16: Can the authors use the CIP data for the flights where the 2DS was not operating? This would be of interest as it would extend the analysis to colder temperatures, where there is arguably larger differences in the INP parameterizations.

CIP data has been added to Fig. 4b (below). This shows that the INP parameterisations overestimate ice crystal concentrations at the lowest sampled temperatures (approximately -27 °C).

[Figure]

*Figure 4. Box and whisker plots summarising in cloud measurements (averaged over 10 s) as a function of temperature. Plate a) shows the concentration of cloud droplets (cm$^{-3}$), measured by CAS, while b) shows the concentration of ice particles measured by 2DS and CIP-25, based on those classified as highly irregular (see text for details). The concentration of ice nucleating particles predicted by the DeMott et al. (2010) parameterisation with a high (1 scm$^3$) and low (0.1 scm$^3$) aerosol input are shown as dark and light blue lines, respectively in b). The green line is the predicted ice particle concentration according to the Cooper (1986) parameterisation. c) a histogram of the flight distance while continuously sampling ice.*

9. Page 13 line 22: Again no clear trend is evident. There are very few data points at temperatures lower than -15 C and there is a lot of scatter in the data.

This refers to temperatures greater than -15C, which is where the majority of measurements during the campaign were made. This sentence has been clarified to read:

"At temperatures greater than -15 °C there is a trend of the ice crystal concentrations showing greater variability and higher median concentrations with increasing temperature."

10. Page 13 line 24: Is the figure showing the histogram missing? This is key to demonstrate that the ice occurs in small patches in the liquid clouds.

This figure has been added to the manuscript (Fig. 4c):

11. Page 15: Why do the authors only sub-select data in the H-M temperature range where secondary ice production might be expected to be enhanced? It would be beneficial to also look to see if there was a relationship outside of this temperature zone e.g. to see if there is any evidence of primary ice being formed from the freezing of drizzle drops (e.g. Rango and Hobbs 1991).

As discussed in the manuscript laboratory experiments have suggested that H-M production rates are proportional to the accumulation of large drops (>24 μm) (Mossop and Hallett, 1974). The aim of Fig. 5 was to examine if such a relationship could be identified in the MAC field measurements. As suggested similar plots have been added to the manuscript for temperatures <-8 C.

[Figure]

*Figure 5a,b. The relationship between the concentration of highly irregular (2DS HI) particles and low irregular particles (2DS LI) (low irregular particles greater than approximately 80 µm). Figures 6c,d and 6e,f show the relationship with the concentration of droplets larger than 30 and 20 µm, respectively. Panels on the left (a, c and e) show measurements at temperatures lower than -8°C and panels on the right (b, d and f) show those in the range -8 to 0 °C. The black lines are the 25th, 50th and 75th percentile of the 2DS HI concentration for each droplet concentration bin.*

12. Page 18 and figure 9: What is the mechanism by which glaciation would occur preferentially in downdrafts? 13. Figure 9: From the figure, it looks like the frequency of downdrafts measured by the aircraft is much larger than the updrafts. This was surprising to me and I would like the authors to confirm that there is no instrumental bias in the vertical

 Assuming that there is no bias, can the authors explain the higher frequency of downdrafts in the measurements?

Given the uncertainty in the vertical wind measurements this has section has been removed from the paper. In anticipated that this will be incorporated into ongoing work examining eddy covariance flux measurements from the aircraft.

14. Page 19: Is the implication that the columns observed at warmer temperatures are generated by secondary ice processes? If so make that clear to the reader and link to figure 5 perhaps.

The paper is structured so that Sect. 3 presents the measurements with minimal interpretation, while Sect 4. provides more subjective discussion of the results. Section 4.2 makes the link between crystal habit and secondary production.

15. Figure 11: Can you include CPC data from the ground site? Also, how is number concentration derived from the aerosol scattering cross-section?

Measurements from the CPC at the ground site have been added to this figure. Particles are sized using Mie scattering theory assuming spherical particles of known refractive index.

16. Page 21 line 13: Presumably any surface generated aerosol is inhibited from being transported above the boundary layer as there are likely to be strong thermodynamic gradients co-incident with the top of single-layer clouds? If the data were normalised and plotted relative to cloud/boundary layer top (see point 4), you may expect to see sharper vertical changes in the profile of aerosol. These would be smeared out in figure 11 as all flight data are simply plotted relative to altitude and the data are therefore averaged over different boundary layer depths. Also, if the boundary layer was not well-mixed, then this could also result in a drop-off in the concentration of surface generated aerosol with altitude. Can you examine the aircraft thermodynamic data to examine if this was important?

The following figures show the aerosol concentrations plotted relative to the cloud top height. The concentration of aerosol in the size range 0.5 to 1.5 µm decrease above cloud top as would be expected from a surface source of aerosol. While the CPC shows layers above cloud with high concentrations. These features are the same as the original plot in the paper where the data was plotted against absolute altitude, so the manuscript is unchanged.

[Figure]

17. Page 23 line 8: Is there any evidence that the enhanced aerosol concentrations observed above cloud are being entrained into the cloud top? The suggested linkage between high aerosol concentrations above cloud top and the presence of clouds is rather tenuous.

The lack of correlation between above cloud aerosol and cloud concentrations across the flights suggests that the entrainment is relatively limited.

18. Page 24 lines 6 to 10: This short paragraph on the WIBS data (which is not very informative) should either be expanded to link to the aircraft measurements or other ground based data, or removed given that a subsequent paper on this data is planned.

Given this section discusses aerosol measurements it is appropriate to provide some information on the aerosol composition. Bioaerosol are particularly important since they have been identified as an important high temperature INP. However, these measurements are

described in detail in a recently published ACPD paper, so are only briefly summarised here. A reference to the ACPD paper has been added to the manuscript.

19. Page 26 line 1: Can you use back-trajectories to see if there is a different source region for the case where the aerosol hygroscopicity increased between 28 and 29 Nov?

HYSPLIT trajectories indicate this airmass had passed over sea ice/open water regions. However after 30 November 2015 the hygroscopicity was relatively consistent and does not show a significant relationship with the wind direction/airmass history. This information has been added to the text.

20. Page 27 line 15: What about using the below-cloud aerosol concentration? This is more likely to be relevant, especially for cases were no elevated aerosol layers were observed to be in contact with cloud top.

No significant correlation was observed between below cloud aerosol and ice concentrations.

21. Figure 14: Is the data in the top right panel suggesting that the source of the elevated aerosol concentrations are from the southern Ocean? Are these above cloud?

Yes it does. Most of the measurements between 3 and 4 km had high clouds above.

22. Figure 15: It is somewhat surprising that given that there is evidence of a correlation between aerosol concentrations and source region (Fig 14) that this is not apparent in cloud drop concentration. I would have expected increases in the aerosol concentrations to result in elevated cloud drop concentrations in these liquid dominated layer clouds.

Yes this might be expected. The strongest relationship between aerosols and airmass history is for particles 0.5 to 1.5 µm this is only a small proportion of the total CCN. While the CPC covers the CCN but also smaller particles. Also, given that the majority of measurements were conducted over broken sea ice, it may be that the CCN origin may be more local and not show up in the far field trajectories. This discussion has been added to the text.

23. The discussion section focuses solely on ice production in the clouds. Given that these are liquid dominated clouds the authors should also include some discussion on the liquid phase.

The following paragraph has been added about the liquid phase:

"This section summarise the observations presented in the paper and discusses the important microphysical processes. The cloud types were generally stratus, both single and multiple layers, predominantly between -20 and -3 °C.  These were dominated by super cooled liquid drops, with a median concentration of 113 cm-3. Droplet concentrations were relatively consistent during the campaign with an inter-quartile range of 86 cm-3. The exceptions to this were when the droplets were depleted by high ice concentrations and also flight 217 where anomalously high droplet concentrations were observed, which was associated with an enhanced aerosol layer below cloud. Similar to Arctic layer clouds (McFarquhar et al., 2007), liquid content and cloud drop effective radius both increased with distance from cloud base likely due to condensational growth. Collision coalescence may also have contributed to increase in effective radius. However, droplet number concentration was relatively invariant to position within the cloud."

24. Page 25 line 27: Can this not be estimated given that the aircraft was doing vertical profiles?

There isn't a line 27 on Page 25. Please could the reviewer clarify what this refers to?

Minor comments and technical corrections
1. Page 1 line 23: Additional clarification on what "key processes" means is required.

This refers to processes such as droplet activation, primary and secondary ice nucleation as well as the impact of dynamics and meteorology. To avoid the abstract becoming overly long these haven't been listed, but are discussed in the introduction and the remainder of the paper.

2. Page 1 line 27: Clarify that the size quoted is particle diameter.

Yes this is correct and clarified in the text.

3. Methods: It would seem more appropriate to introduce the NAME model in the methods section instead of in the results section.

This paragraph has been moved as suggested.

4. Page 6 line 22: I don't think the CAS part of a CAPS probe has anti-shatter tips. Please check. Also, was the CDP fitted with anti-shatter tips?

The CDP does not have anti-shatter tips. The leading edge of the CAS is fairly sharp and would have some anti-shatter properties, but we have removed the description of this as "anti-shatter" in the paper.

5. Are the aircraft altitudes relative to mean sea level?

Yes. This has been clarified in the text.

6. Page 6 line 28: Can you put a measure of uncertainty on your IWC estimate that results from assuming the Brown and Francis mass-diameter relation?

The Brown & Francis mass-diameter relation has been used by a number of previous mixed phase cloud studies (e.g. Lloyd et al., 2015, Crosier et al., 2011). However, several other M-D relationships exist. To estimate the uncertainty in Brown & Francis we re-calculate the IWC using the Heymsfield et al. (2004) M-D relationships. Differences between the two relationships are found to be approximately 20%.

Heymsfield, A. J., Bansemer, A., Schmitt, C., Twohy, C., and Poellot, M. R.: Effective ice particle densities derived from aircraft data, J. Atmos. Sci., 61, 982–1003, doi:10.1175/1520-0469(2004)061<0982:eipddf>2.0.co;2, 2004.

7. Page 8 line 5: The MI data is only shown in one small subset of data (Fig 8).

This line has been removed.

8. Figure 2: Is this a flight average or an in-cloud average PSD? Rather than showing one PSD, can you give a more quantitative measure of the agreement between instruments over all flights e.g. compare drop number and LWC from CDP/CAS, IWC from 2DS/CIP, aerosol concentration from GRIMM/CAS. It is also worth describing why you choose certain probes in your analysis e.g. CAS appears to be used preferentially for LWC over the CDP.

This figure has been removed as suggested. The regression equations for all CAS/CDP and CIP/2DS measurements for the overlapping size ranges have now been included in the manuscript. They are as follows:

CDP = $0.87 \times$ CAS + 1.7 cm$^{-3}$ ($R^2$ = 0.83) and CIP = $0.65 \times$ 2DS + 0.7 cm$^{-3}$ ($R^2$ = 0.34)

A comparison between all CAS and GRIMM measurements can be found in Sect. 3.2. The CAS and 2DS were preferentially used over the CDP and CIP due to their higher resolution size bins.

9. Page 9 line 10 to 14: Suggest removing if data is not shown/discussed.

The paper describing these measurements is now in review in ACPD. This reference has been added to the paper. These results are also briefly presented in Section 3.2.

10. Figure 4: What is the point of figure 4b? Is it just to zoom in on the data closer to the surface? If so, why cut-off the x-axis at 0.3 when there are many data points above this as shown in figure 4a. Consider removing the lower panel.

The lower panel has been removed.

11. Figure 5: The continuous colorbar used does not adequately discriminate different flights. Suggest using an alternative way of plotting each flight or remove the colorbar.

The colour scale has been removed.

12. Page 12 line 10: The authors state the reason is not clear but then go on to say that higher aerosol concentrations were observed which could explain the higher drop numbers. Do you see any clear correlation between sub-cloud aerosol and cloud drop number across the different flights?

Cloud droplet number and below cloud CPC concentrations were not strongly correlated across the different flights ($R^2$ = 0.22).

13. Figure 10: The figure caption needs to include a lot more information.

A description of the image scale and times have been added to the caption.

14. Page 18 line 10: Replace 3:58 with 15:58 and 4:04 with 16:04 to be consistent with figure axis label.

Done.

15. Throughout the manuscript the authors use the abbreviation "ca". Why not simply write "circa"? Or consider replacing with "approximately" or "about", both of which are used in this context more extensively in the English language.

Done.

16. The authors switch between using cm-3 and scm-3 when referring to number concentrations of cloud and aerosol particles. Cloud drop number concentration data are shown in cm-3 (Figs 5, 6, 8, 12, 15), whereas aerosol number concentration data are mostly shown in scm-3 (Figs 11, 12, 14), except for CCN concentrations in Fig 13 which use cm-3. I would suggest being consistent in the use of units throughout andusing cm-3. It is the ambient aerosol number concentration on which the cloud drops form (not the value at STP) that is important for cloud drop number concentration for example.

When assessing whether any correlations exist between cloud droplet/ice concentrations, aerosol concentrations were converted from STP to concentrations at the same temperature and pressure as the cloud. In the figures we have left the concentrations in STP units so that comparisons can be more easily made with other studies in the literature.

Referee 2

The manuscript summarizes the characteristics of the Microphysics of Antarctic Clouds, or MAC campaign conducted in 2015 over coastal Antarctica and the nearby ocean regions. The measurements, comprising cloud and aerosol retrievals from both ground-based and airborne measurements, provide a compelling source of information for this region, that will most likely be of high interest to the atmospheric modelling community for example. The Authors go further into analyzing the key features in the cloud and aerosol retrievals and the processes affecting them. Interesting features are revealed about the cloud ice mass fraction, ice particle types and the aerosol, even though conclusive explanations for many of the observed cloud features seem elusive. However, much of this can be rightfully attributed to the extremely challenging conditions as well as the so far quite low number of observations in this region. The descriptions of the instruments and retrieval techniques, as well as the presentation of data are for the most part adequate and very good.

My main concerns about the manuscript are related to the description of the surrounding conditions during these measurements. I do think the manuscript would greatly benefit from a bit more systematic description of the meteorological conditions as well as the structure of the clouds sampled during the campaign. This could potentially help with the interpretation of the results, many of which are now based simply on microphysical retrievals put together from all available flights. Below, I will try to summarize these points with more specific comments, followed by minor and technical comments.

1. The manuscript does not provide very detailed information about the height and depth of the sampled cloud layers. If there are considerable differences in the altitude and depth of the cloud layer, this would imply differences in the cloud dynamics as well as in the large-scale meteorological setting from one flight to the next, and would therefore be worth a look to support the subsequent analysis. It is also rather difficult to follow whether the analyzed data (in general but also in the few flight specific examples) represent the cloud base, in-cloud or cloud top conditions. For example, the vertical profile of ice mass fraction given in Fig 4 is interesting information, but it would make it far more interesting, if that data could be put in the context of sampling level with respect to the vertical extent of the sampled cloud.

A table (Table 1) has been added giving the temperature and height of cloud base/top. Figure 3 now shows how cloud properties vary relative to cloud base/top for single layer clouds. Please also see response to reviewer 1 comment 4 where this is discussed further.

2. Would it be possible to consider the existence of multiple cloud layers for all of the flights? It is commented in Section 4.1 that seeding effects were not detected apart from the frontal cloud case. However, could the possibly overlapping cloud layers affect the radiation budget of the sampled clouds, that might impact mixing and perhaps entrainment and thus the cloud properties?

Table 1 has been added to the manuscript and gives a description of whether multiple cloud layers were present. Discussing the cloud radiation budget in sufficient detail to be useful would likely make the paper overly long. It is anticipated that this will discussed in a separate paper where the radiation measurements will be compared with modelled values.

3. While the manuscript does consider the impact of airmass history on particle number concentrations in Section 3.5, it does not clearly outline how these shifts affect other aspects, such as large-scale meteorological forcing, which could impact the cloud properties. Generally, I think it would add great value to put these observations into context by including some more detailed information about the meteorological conditions during the campaign.

The information on the meteorological setting has been extended, we have included surface pressure charts and HYSPLIT back trajectories for each flight in the supplementary material. A table has been added to the manuscript (Table 1) detailing cloud base/top height and temperatures. Also included is a description of whether multiple cloud layers were present.

4. Related to the above, it would be interesting to couple meteorological data (e.g. wind speed) with estimates of the cloud altitude (as per the comments no. 1) in order to estimate the potential of blowing snow to affect the measured cloud properties. This possibility is considered in Section 4.1 but explicitly only for one flight. A more detailed evaluation of the prevailing conditions would allow a broader consideration of at least the possibility of blowing snow contribution also for other flights.

The figure below shows histograms of the horizontal wind speed at different altitudes for measurements in cloud where ice is (blue lines) and is not (red lines) present. No strong relationship can be identified between the wind speed and presence of ice in the clouds. Though it should be noted that a high proportion of measurements are at wind speeds greater than 7 m s$^{-1}$, which is often considered a threshold for blowing snow.

[Figure]

*Figure. Histograms of the horizontal wind speed at different altitudes for measurements in cloud where ice is (blue lines) and is not (red lines) present.*

The following text has been added to the manuscript discussing the relationship between surface type and the presence of ice precipitation:

"Evaluating the impact of these mechanisms during MAC is challenging since most of the in-cloud sampling was performed over snow covered sea ice, making it difficult to attribute local differences in the microphysics to the surface type. Figure 14 shows histograms of the surface albedo for out-of-cloud measurements (below 100 m) when there was (blue line) and was not (red line) ice observed. Here the surface albedo is used as a proxy for the surface type, since values near 0 correspond to overflying open water and the values near 1 correspond to a snow/ice covered surface. Figure 14 suggests that ice measured by the

aircraft while out cloud almost exclusively occurred when overflying a snow/ice covered surface, implying a link between the surface type and the presence of ice in the clouds. The ice measured on the aircraft when out-of-cloud could either have originated from the surface or precipitated from clouds above. However, it should be noted that very few measurements were made over open water regions.

[Figure]

*Figure 14. Histograms of the surface albedo for of out-of-cloud measurements (below 100 m) when there was (blue line) and was not (red line) ice detected."*

Other minor and technical comments

1. The notation scm-3 is used throughout the paper for the units of aerosol concentration, apparently referring to concentration defined in STP conditions, yet for cloud droplets the more commonplace cm-3 is used. I think the STP-related notation should be clearly defined and it should be made very clear where each of the notations are used to avoid confusion. However, I do think the best option by far would be to just use the same units (cm-3) for all concentrations.

We have clarified that this refers to standard temperature and pressure. In the figures we have left the concentrations in STP units so that comparisons can be more easily made with other studies in the literature.

2. Page 7, line 9: Please add a short definition for the circularity.

Done.

3. Fig 4, "Radar altitude" vs "altitude", are the panels from different datasets? Please explain.

The radar altitude gives the height above the ground and "altitude" is above mean sea level (this is clarified in the text). However the trends are the same for both plots. For this reason, the radar altitude plot has been replaced with a plot showing normalised position within the cloud (please see our response to reviewer 1 comment 4)

4. Page 13 lines 24-27: Histogram of the spatial extent of ice: which figure should I be looking at here? If it is missing, please consider adding a figure showing this.

This figure has been added to the manuscript (Fig 4c).

5. Page 15, and fig 6: Why do you limit the analysis to the temperature range -8...-3 in this particular case? Please elaborate.

Please see our response to reviewer 1 comment 11, which answers this question.

6. The terminology regarding Fig 7 is confusing: page 16, line 9 "particle size distributions", page 17, line 2 "droplet spectrum" and just "size distributions" in the caption. Please try to use more consistent terminology here so it's easier to read and to know exactly what you refer to.

"Particles size distribution" is now used throughout.

7. Fig 7: You don't mention the dashed lines in the caption or elsewhere in the text. Please add an explanation for these or remove them from the figure.

Dashed lines show measurements from the CAS and solid lines are from the 2DS. This is now clarified in the text.

8. Page 17, line 5: Could you please add some numbers to make a more clear case for the droplet depletion?

Concentrations have been added to the manuscript.

9. Page 19: Please consider adding a dedicated subsection for the analysis of the cloud particle images. Coming directly after the quantitative results on ice fractions and number concentrations it feels a little out of place to me.

As suggested this has been given its own sub section.